# SARS-CoV-2 remodels the Golgi apparatus to facilitate viral assembly and secretion

Jianchao Zhang[1,2]*, Andrew Kennedy[3], Daniel Macedo de Melo Jorge[4], Lijuan Xing[1], Whitney Reid[1], Sarah Bui[1], Joseph Joppich[1], Molly Rose[1], Sevval Ercan[1], Qiyi Tang[5], David Ginsburg[2,6], Andrew W. Tai[3,4]*, Yanzhuang Wang[1,7]*

1 Department of Molecular, Cellular and Developmental Biology, University of Michigan, Ann Arbor, Michigan, United States of America, 2 Life Sciences Institute, University of Michigan, Ann Arbor, Michigan, United States of America, 3 Department of Internal Medicine, University of Michigan Medical School, Ann Arbor, Michigan, United States of America, 4 Department of Microbiology and Immunology, University of Michigan Medical School, Ann Arbor, Michigan, United States of America, 5 Department of Microbiology, Howard University College of Medicine, Washington, DC, United States of America, 6 Departmentof Internal Medicine, Human Genetics, and Pediatrics, University of Michigan, Ann Arbor, Michigan, United States of America, 7 Institute of Neurological and Psychiatric Disorders, Shenzhen Bay Laboratory, Shenzhen, Guangdong, China

☯ These authors contributed equally to this work.
* jczha@umich.edu (JZ); andrewwt@med.umich.edu (AWT); yzwang@szbl.ac.cn (YW)

## Abstract

The COVID-19 pandemic is caused by the enveloped virus SARS-CoV-2. Despite extensive investigation, the molecular mechanisms for its assembly and secretion remain largely elusive. Here, we show that SARS-CoV-2 infection induces global alterations of the host endomembrane system, including dramatic Golgi fragmentation. SARS-CoV-2 virions are enriched in the fragmented Golgi. Blocking endoplasmic reticulum (ER) to Golgi trafficking dramatically inhibits SARS-CoV-2 assembly and secretion without reducing viral genome replication. Significantly, SARS-CoV-2 infection down-regulates GRASP55 but up-regulates TGN46 protein levels. Surprisingly, GRASP55 expression reduces both viral secretion and spike number on each virion without affecting viral entry, while GRASP55 depletion displays opposite effects. In contrast, TGN46 depletion only inhibits viral secretion without affecting spike incorporation into virions. Taken together, we show that SARS-CoV-2 alters Golgi structure and function to modulate viral assembly and secretion, highlighting the Golgi as a potential therapeutic target for blocking SARS-CoV-2 infection.

## Author summary

SARS-CoV-2, the etiologic agent of COVID-19, is an enveloped RNA virus. Despite extensive research, the molecular mechanisms governing SARS-CoV-2 assembly and secretion remain poorly understood. While some studies suggest that SARS-CoV-2 bypasses the Golgi apparatus via an unconventional secretory

**Data availability statement:** All relevant data are within the manuscript and its Supporting Information files.

**Funding:** This work was supported by the National Institute of Allergy and Infectious Disease R21AI152865 (A.W.T.), National Institute of General Medical Sciences R01GM139823 (A.W.T.) and R35GM130331 (Y.W.), National Institute of Neurological Disorders and Stroke R01NS102279 (Y.W.), National Institutes of Health National Heart, Lung, and Blood Institute R35HL171421(D.G.), the American Lung Association Grant COVID-1019544 (Y.W. and A.W.T.), and the Fast Forward Protein Folding Disease Initiative of the University of Michigan (A.W.T. and Y.W.). The funders had no role in study design, data collection and analysis, decision to publish, or preparation of the manuscript.

**Competing interests:** The authors have declared that no competing interests exist.

pathway, we provide compelling evidence that the virus instead traffics through the Golgi and relies on the conventional ER-to-Golgi pathway for virion assembly and release. Intriguingly, SARS-CoV-2 infection down-regulates the Golgi stack protein GRASP55 while up-regulates TGN46, a less characterized trans-Golgi network (TGN) protein. Depletion of GRASP55 enhances both spike protein incorporation into virions and viral trafficking, whereas GRASP55 expression has the opposite effect. In contrast, TGN46 depletion inhibits viral trafficking but not assembly, suggesting distinct roles for these proteins at different stages of infection. Collectively, our findings demonstrate that SARS-CoV-2 modulates GRASP55 and TGN46 levels to remodel the Golgi apparatus, facilitating viral assembly and secretion. This mechanism appears conserved across SARS-CoV-2 variants, advancing our understanding of how the virus exploits the host secretory pathway.

## Introduction

Since December 2019, the unprecedented coronavirus disease 2019 (COVID-19) pandemic caused substantial morbidity and mortality worldwide. SARS-CoV-2 (severe acute respiratory syndrome coronavirus 2), the etiologic agent for COVID-19, is an enveloped, single-stranded positive-sense RNA virus in the beta-coronavirus family. Vaccination is the most critical and effective strategy to prevent severe COVID-19 disease. However, vaccines are not effective in all individuals, such as the immunocompromised, and the possible emergence of SARS-CoV-2 variants with immune evasion and/or increased pathogenicity continues to pose a threat to existing vaccines [1]. Therefore, it remains important to elucidate the detailed mechanisms of the SARS-CoV-2 infection cycle in order to develop strategies to block the replication and spread of all variants.

Entry into host cells is the first step of virus infection and acts as a determinant of host susceptibility and viral pathogenesis. Like SARS-CoV, SARS-CoV-2 employs human angiotensin-converting enzyme 2 (ACE2) as the entry receptor. The spike (S) protein on the virus interacts with ACE2 at the host cell surface, allowing the virus to attach to the host cell membrane. After cleavage by the transmembrane serine protease 2 (TMPRSS2), the spike protein is activated and initiates fusion of the viral membrane with the host cell plasma membrane. When TMPRSS2 is absent, SARS-CoV-2 is endocytosed and the spike protein is activated by Cathepsin B and L (Cat-B/L) inside the lysosomal lumen [2]. Compared to SARS-CoV, the spike protein of SARS-CoV-2 possesses a unique furin cleavage site that facilitates its cell entry [3]. Following entry, the genomic RNA is released into the cytoplasm. The polyproteins pp1a and pp1ab, translated from two large open reading frames ORF1a and ORF1b, are self-processed into 16 individual non-structural proteins (nsps), which modulate the endoplasmic reticulum (ER) to form double-membrane vesicles (DMVs) where the viral replication and transcription complex (RTC) is anchored to produce abundant viral RNAs [4].

In addition to the 16 nsps, the SARS-CoV-2 genome also encodes four structural (S, M, E, and N) and 9 accessory proteins [5]. Three of the four viral structural proteins (S, E, and M) are synthesized in the ER and transported to the ER-to-Golgi intermediate compartment (ERGIC), where virions are assembled and budded into the lumen of the ERGIC and Golgi. In the ER and Golgi, the spike protein is highly glycosylated [6,7], which affects its folding, stability, interaction with ACE2, and susceptibility to vaccines. Integrative imaging revealed that SARS-CoV-2 infection triggers Golgi fragmentation [8], but the cause and consequence of Golgi fragmentation remain largely elusive.

Mature viral particles of several coronaviruses, including SARS-CoV, are generally believed to traffic through the conventional protein secretory pathway for release [9–11]. However, it has been proposed that SARS-CoV-2 exits host cells in a Golgi-independent manner by lysosomal exocytosis in 2020 [12,13] supported by a number of following studies [14–18], or by small secretory vesicles likely from an unconventional secretory pathway [19]. Thus, it is widely accepted that SARS-CoV-2 undergoes a Golgi-bypass unconventional pathway for release, stated by numerous reviews [20–27].

Here, we systematically investigated the interaction between SARS-CoV-2 and the host endomembrane system. Our study revealed that SARS-CoV-2 infection causes a dramatic disruption of the Golgi structure. Furthermore, a targeted small molecule screen demonstrates that disruption of ER to Golgi trafficking significantly reduces viral assembly and secretion. By surveying a large number of Golgi proteins, we found that SARS-CoV-2 infection reduces GRASP55 but increases TGN46 protein level. Importantly, expression of GRASP55 or depletion of TGN46 in cells significantly inhibits viral infection and secretion. Interestingly, manipulation of GRASP55 also affects spike number on each virion. Therefore, our study uncovered a novel mechanism by which SARS-CoV-2 remodels the Golgi structure and function to facilitate viral assembly and secretion, which supports the secretory pathway model of SARS-CoV-2 release and identifies potential novel drug targets to interrupt the SARS-CoV-2 infection cycle.

## Results

### SARS-CoV-2 infection induces dramatic morphological alterations of multiple subcellular organelles, in particular the Golgi

Enveloped viruses often utilize the host secretory pathway to assemble and exit from the cell [28]. How SARS-CoV-2, an enveloped beta-coronavirus, modulates the endomembrane system for its assembly and release is an important but unanswered question. To address this question, we performed morphological analyses of multiple subcellular organelles, including the ER, ERGIC, Golgi, autophagosome, endosome, and lysosome, in SARS-CoV-2-infected Huh7-ACE2 cells, a hepatoma cell line that is often used in studies of SARS-CoV-2 infection [29]. After 24 h infection by SARS-CoV-2 (USA-WA1 strain, same as below unless specified), the ER structure marked by calreticulin was not dramatically altered (Table 1, S1A-C Fig). ERGIC53 was normally concentrated in the perinuclear region in uninfected cells, while in SARS-CoV-2 infected cells it displayed as dispersed puncta distributed in the cytoplasm (Fig 1A-B). Further quantitation of the images indicated that the total area of ERGIC53 did not change (Fig 1C), and its relative expression level was modestly reduced (Fig 1D).

The Golgi structure was significantly affected by SARS-CoV-2 infection. In contrast to the well-organized ribbon-like structure in uninfected cells, as labeled by the Golgi protein GRASP65, the Golgi was seen as small fragments dispersed in the entire cytoplasm in infected cells (Fig 1E-F), although the total area and expression level of GRASP65 did not change (Fig 1G-H). In control cells, COPI, COPII, and clathrin coats, indicated by β-COP, Sec31, and clathrin heavy chain, respectively, are accumulated in the perinuclear region, whereas in SARS-CoV-2 infected cells they are also dispersed in the cytoplasm (S1D-O Fig), consistent with Golgi fragmentation.

The endosome/lysosome system was also affected by SARS-CoV-2 infection. The early endosome marker EEA1 and the late endosome marker Rab7 were both found in larger puncta in infected cells (Fig 1I-P), suggesting aggregation of the membranes. While EEA1 expression level did not change (Fig 1L), Rab7 was up-regulated and showed an increase

**Table 1. Summary of morphological changes of subcellular organelles after SARS-CoV-2 infection.**

| Subcellular localization | Marker | Morph. alteration | Item # per cell | | | Area per cell | | | Relative Expression level | | |
|---|---|---|---|---|---|---|---|---|---|---|---|
| | | | uninfected | infected | p-value | uninfected | infected | p-value | uninfected | infected | p-value |
| ER | Calreticulin | – | NA | | | 671.8±159.6 | 613.0±204.9 | n.s. | 1.00±0.30 | 0.93±0.34 | n.s. |
| ERGIC | ERGIC53 | Dispersed | 131.6±21.2 | 95.1±24.4 | *** | 99.9±22.5 | 97.4±28.2 | n.s. | 1.00±0.20 | 0.85±0.32 | * |
| Cis-Golgi | GM130 | Dispersed | 15.9±5.7 | 70.8±26.3 | *** | 68.6±13.0 | 73.3±45.3 | n.s. | 1.00±0.19 | 0.78±0.47 | * |
| | GPP130 | Dispersed | 13.5±5.3 | 54.1±21.9 | *** | 51.8±19.3 | 59.2±27.2 | n.s. | 1.00±0.28 | 1.07±0.39 | n.s. |
| | GRASP65 | Dispersed | 11.7±4.2 | 79.2±32.7 | *** | 54.3±9.7 | 55.6±17.2 | n.s. | 1.00±0.24 | 0.87±0.31 | n.s. |
| Medial/ Trans-Golgi | GRASP55 | Dispersed | 16.8±7.2 | 41.2±10.2 | *** | 69.3±15.3 | 34.2±25.5 | *** | 1.00±0.24 | 0.32±0.24 | *** |
| Trans-Golgi | GCC88 | Dispersed | 18.8±8.0 | 79.1±30.1 | *** | 49.3±12.9 | 43.9±16.6 | n.s. | 1.00±0.15 | 0.79±0.30 | *** |
| | GalT | Dispersed | 20.1±6.2 | 54.4±21.4 | *** | 41.2±10.0 | 44.3±19.1 | n.s. | 1.00±0.27 | 0.94±0.41 | n.s. |
| Trans-Golgi network | TGN46 | Dispersed | 25.3±9.5 | 89.5±39.6 | *** | 33.5±11.3 | 93.0±40.2. | *** | 1.00±0.33 | 2.38±1.08 | *** |
| | Golgin-245 | Dispersed | 11.6±4.2 | 61.5±22.7 | *** | 53.2±8.8 | 49.6±20.9 | n.s. | 1.00±0.29 | 0.64±0.16 | *** |
| | Arl1 | Dispersed | 12.0±5.0 | 51.1±19.1 | *** | 58.8±19.1 | 56.4±17.4 | n.s. | 1.00±0.24 | 0.83±0.39 | *** |
| COP-I | β-COP | Large granules | 112.6±19.3 | 79.8±21.8 | *** | 148.5±34.7 | 152.8±46.4 | n.s. | 1.00±0.16 | 0.86±0.34 | * |
| COP-II | Sec31 | Large granules | 142.2±4.2 | 115.7±2.7 | *** | 56.6±3.6 | 48.1±3.5 | n.s. | 1.00±0.04 | 0.72±0.03 | *** |
| Clathrin-coated vesicles | Clathrin heavy chain | Dispersed | 66.5±25.5 | 65.0±28.8 | n.s. | 40.7±32.1 | 34.9±25.5 | n.s. | 1.00±0.44 | 0.90±0.57 | n.s. |
| Endosome | EEA1 | Large granules | 62.3±16.3 | 31.2±13.7 | *** | 19.3±6.1 | 19.1±8.8 | n.s. | 1.00±0.30 | 1.14±0.70 | n.s. |
| | Rab7 | Large granules | 111.2±26.1 | 118.8±40.4 | n.s. | 22.3±5.8 | 50.1±24.2 | *** | 1.00±0.26 | 2.24±1.02 | *** |
| Lysosome | CatD | Larger granules | 126.7±40.2 | 106.4±33.5 | * | 79.8±37.3 | 69.0±34.4 | n.s. | 1.00±0.37 | 0.90±0.39 | n.s. |
| | LAMP2 | Large granules | 128.1±24.7 | 99.5±26.9 | *** | 143.1±34.9 | 146.3±68.9 | n.s. | 1.00±0.25 | 0.97±0.42 | n.s. |
| Cytoskel-eton | Tubulin | – | NA | | | 956.6±262.9 | 784.5±258.5 | *** | 1.00±0.26 | 0.91±0.48 | n.s. |
| | Actin | – | NA | | | 472.9±124.3 | 501.4±120.0 | n.s. | 1.00±0.28 | 1.08±0.29 | n.s. |

All quantitation data are shown as mean±SD from three independent experiments. Statistical analyses were performed using two-tailed Student's t-test. *, $p < 0.05$; ***, $p < 0.001$; n.s., not significant.

in the area of their vesicles (Fig 1O-P). The aggregation of endosome/lysosome membranes was supported by the late endosome/lysosome marker LAMP2, which displayed fewer but larger puncta in infected cells (Fig 1Q-T). In addition, modest effects were observed for LC3 and Cathepsin D (S2A-H Fig), which represent autophagosome and active lysosomes, respectively. While microtubule cytoskeleton appeared to be partially depolymerized, actin microfilament organization did not seem to change in SARS-CoV-2 infected cells (S2I-N Fig). SARS-CoV-2 infection of Huh7-ACE2 cells at MOI from 1 to 3 for 24 h did not induce significant cell death (S2O Fig). Taken together, SARS-CoV-2 infection alters the structures of the ERGIC, endosomes, lysosomes, and particularly, the Golgi.

### Disrupting Golgi functions by small molecules inhibits SARS-CoV-2 infection

We reasoned that the morphological changes of the endomembrane system may facilitate SARS-CoV-2 infection, and if so, disrupting the functions of these membrane organelles may impact viral infection. Therefore, we performed a focused

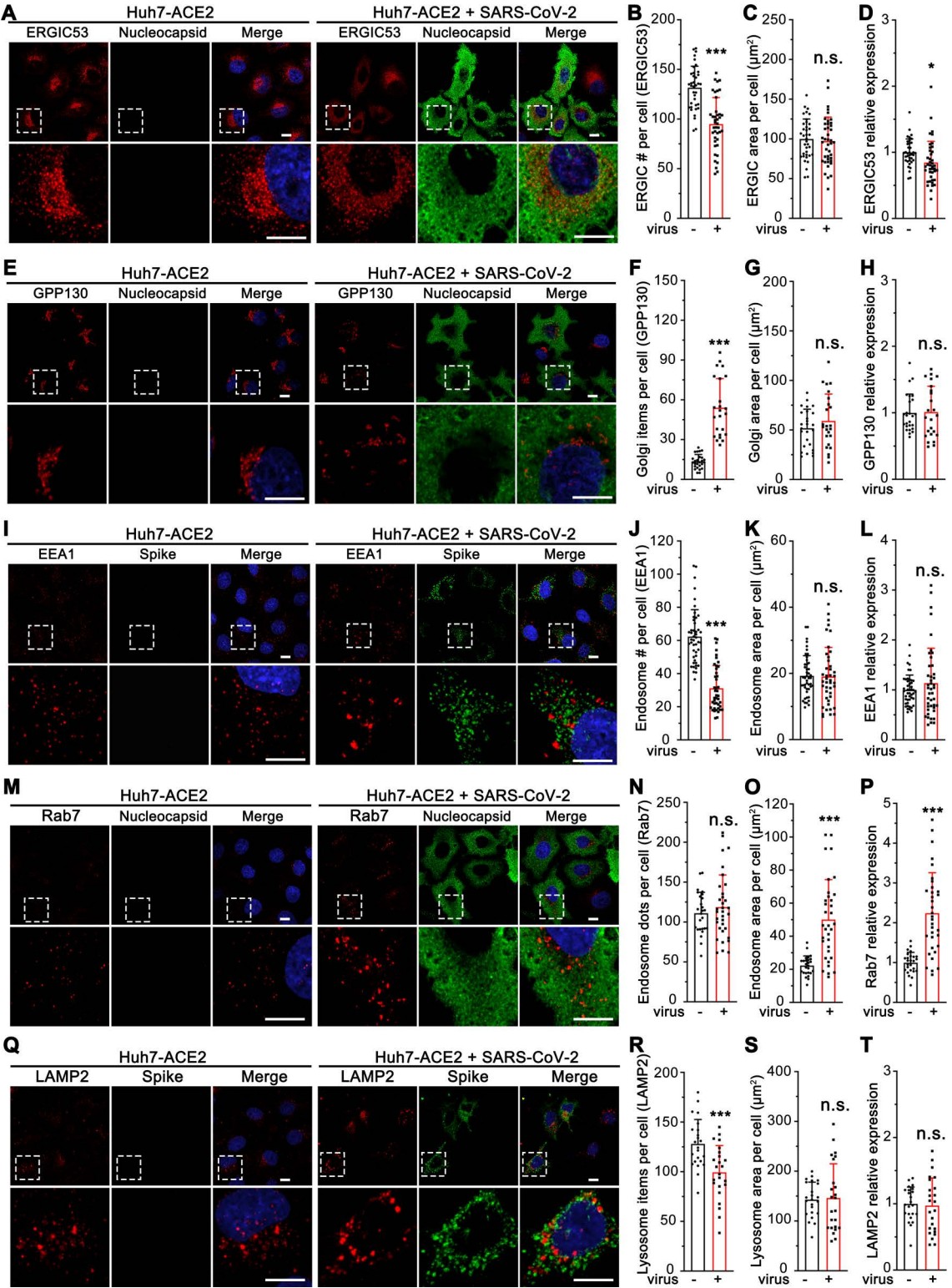

**Fig 1. SARS-CoV-2 infection induces global morphological changes of host cell membrane organelles.** (A) Representative confocal images of Huh7-ACE2 cells incubated with or without SARS-CoV-2 (MOI = 1) for 24 h and stained for ERGIC53 and nucleocapsid. (B-D) Quantification of ERGIC53 for the number of puncta (B), area (C), and relative expression (D) in A. (E) Representative confocal images of Huh7-ACE2 cells incubated with or

without SARS-CoV-2 (MOI = 1) for 24 h and stained for a cis-Golgi marker GPP130 and nucleocapsid. (F-H) Quantification of GPP130 in E. (I) Representative confocal images of Huh7-ACE2 cells incubated with or without SARS-CoV-2 (MOI = 1) for 24 h and stained for an early endosome marker EEA1 and spike. (J-L) Quantification of EEA1 item number (J), area (K), and relative expression (L) in I. (M) Representative confocal images of Huh7-ACE2 cells incubated with or without SARS-CoV-2 (MOI = 1) for 24 h and stained for a late endosome marker Rab7 and nucleocapsid. (N-P) Quantification of Rab7 in M. (Q) Representative confocal images of Huh7-ACE2 cells incubated with or without SARS-CoV-2 (MOI = 1) for 24 h and stained for a late endosome/lysosome marker LAMP2 and spike. (R-T) Quantification of LAMP2 in Q. Boxed areas in the upper panels are enlarged and shown underneath. Scale bars in all panels, 10 μm. All quantitation data are shown as mean ± SD from three independent experiments. Statistical analyses are performed using two-tailed Student's t-test. *, $p < 0.05$; **, $p < 0.01$; ***, $p < 0.001$; n.s., not significant.

small molecule screen to determine whether molecules targeting the endomembrane system could inhibit SARS-CoV-2 infection. Prior to the experiment, we titrated each chemical compound and selected the highest dose that was widely used and did not cause detectable cell death. We then used this concentration to treat cells during SARS-CoV-2 infection. Among a total of 27 drugs tested, 22 drugs displayed significant inhibitory effects on SARS-CoV-2 infection (Fig 2A-B). Three ER stress inducers, thapsigargin (Tg, a SERCA calcium pump inhibitor), tunicamycin (Tm, a protein glycosylation inhibitor), and dithiothreitol (DTT, a reducing agent that disrupts disulfide bonds of proteins), significantly inhibited viral infection. Interestingly, distinct from Tg, ionomycin (Iono), a calcium ionophore that triggers calcium influx across the plasma membrane as well as ER membranes, exhibited no effect on viral infection. Two Golgi function modulators, brefeldin A (BFA, a fungal product that inhibits ER-to-Golgi trafficking) and monensin (an ionophore that inhibits trans-Golgi transport), displayed over 98% inhibition of viral infection, suggesting an important role of the Golgi in SARS-CoV-2 infection.

## Chemicals interfering with lysosome functions and protein homeostasis inhibit SARS-CoV-2 infection

The lysosome is a key subcellular organelle in both endocytic and exocytic pathways essential for viral entry and secretion. Indeed, disruption of lysosomal function with either bafilomycin A1 (BafA1, an inhibitor of vacuolar-type H+-ATPase that blocks autolysosome acidification), chloroquine (CQ, an FDA-approved antimalarial drug that increases the pH of acidic vesicles), Vacuolin-1 (a cell-permeable compound that prevents lysosomes from fusing with the plasma membrane), a cocktail of protease inhibitors (PIs, which inhibit a broad range of cellular proteases), or three lysosomal protease inhibitors (E64d, leupeptin, and pepstatin), significantly reduced viral infection (Fig 2A-B). Interestingly, both an autophagy inducer (Torin-1) and autophagy inhibitors, 3-methyladenine (3-MA) and CID 1067700 (CID), displayed similar inhibitory effects on viral infection, although to a lesser degree than lysosomal inhibitors, implying that autophagy may also be involved in SARS-CoV-2 infection as previously suggested [30].

The proteostasis network is often hijacked by viruses to satisfy the high demands for viral protein synthesis and folding [31]. Therefore, we examined the effects of blocking different proteostasis pathways on SARS-CoV-2 infection (Fig 2A-B). Inhibition of proteasome-mediated protein degradation with MG132 potently inhibited viral infection, while inhibition of ubiquitin conjugation by PYR-41 (an E1 ubiquitin-activating enzyme inhibitor) displayed no effect. It is possible that a higher concentration of PYR-41 is required to entirely block the first step of ubiquitination; alternatively, the virus may possess the capability to bypass the E1 enzyme for degradation like some bacteria [32]. Inhibition of the AAA ATPase p97/VCP with CB-5083, UPCDC-30245, or NMS-873 also significantly reduced viral infection, indicating that the ER-associated protein degradation (ERAD) pathway may serve as a potential target for the treatment of COVID-19. In addition, three protease inhibitors, PF-429242 (a site-1 protease inhibitor), Decanoyl-RVKR-CMK (a furin inhibitor), and camostat mesylate (a TMPRSS2 inhibitor), displayed modest inhibitory effects on SARS-CoV-2 infection, while aprotinin (an inhibitor of serine proteases including trypsin, chymotrypsin, and plasmin) had no effect. The relative smaller effect of camostat mesylate observed here, compared to previous studies [33], might be due to the use of different cell lines across studies.

Latrunculin B (LatB, an actin polymerization inhibitor) did not inhibit viral infection, but nocodazole (Noc, a microtubule polymerization inhibitor) exhibited about 50% inhibition of viral infection (Fig 2A-B), consistent with a previous report

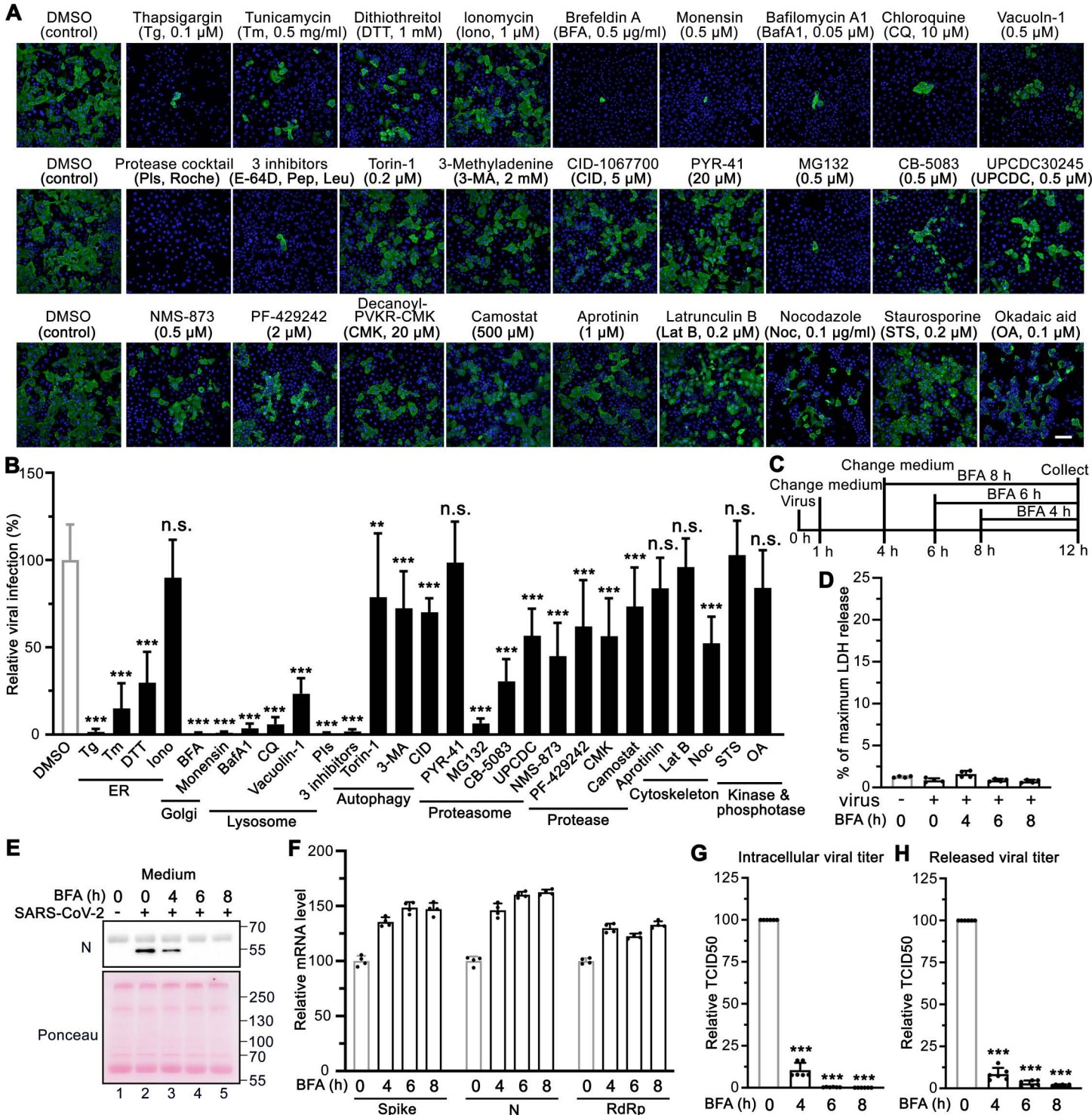

**Fig 2. Chemicals disrupting Golgi functions inhibit SARS-CoV-2 infection.** (A) Representative immunofluorescence images of Huh7-ACE2 cells that were infected with SARS-CoV-2 (MOI = 1) in the presence of indicated chemicals for 24 h and stained for nucleocapsid and DNA. The concentrations of the chemical are shown on the figure or as follows: protease inhibitor cocktails (PIs), 100 ml/tablet; 3 inhibitors, 10 μM E-64D, 20 μM pepstatin, 100 μM leupeptin. Scale bar, 100 μm. (B) Quantification of the viral infection percentage in A, with the control normalized to 100%. Data are shown as mean ± SD from 20-30 random images from two or three independent experiments. Statistical analyses are performed using one-way ANOVA, Tukey's

multiple comparison test. *, p < 0.05; ***, p < 0.001; n.s., not significant. (C) Schematic diagram of the multiple time points for BFA treatment assay. (D) Representative LDH assay of Huh7-ACE2 cells treated with 5 µg/ml BFA for the indicated time points with 4 technical replicates from two independent experiments. (E) Immunoblots of culture media of Huh7-ACE2 cells incubated with or without SARS-CoV-2 (MOI = 5) for 12 h and treated by 5 µg/ml BFA for the indicated time points. (F-H) Representative RT-qPCR assay (F), intracellular viral titer assay (G), and released viral titer assay (H) of Huh7-ACE2 cells treated by 5 µg/ml BFA for the indicated time points with 4 technical replicates from two independent experiments.

that the microtubule cytoskeleton is required for SARS-CoV-2 infection [8]. In our study, neither staurosporine (STS, a broad-spectrum protein kinase inhibitor) nor okadaic acid (OA, an inhibitor of several serine/threonine phosphatases) showed a significant effect on viral infection (Fig 2A-B).

To confirm these observations, we selected the 9 most effective inhibitors, which target the ER (Tg), Golgi (BFA and monensin), lysosome (BafA1, CQ, Vacuolin-1, and 3-inhibitors), and proteostasis (MG132 and CB-5083), respectively, and performed dose-response assays on SARS-CoV-2 infection. All 9 compounds, in particular, Tg, BFA, monensin, BafA1, and 3-inhibitors, consistently inhibited SARS-CoV-2 infection even at much lower concentrations (S3A-B Fig). These 9 compounds did not induce significant cell death at the concentrations used in Fig 2A (S3C Fig). To test whether these molecules inhibit infection at the level of viral entry, we conducted a SARS-CoV-2 spike pseudotyped virus entry assay [34]. As expected, pseudovirus cell entry depends on ACE2 expression (S3D Fig), and is inhibited by BafA1, CQ, and lysosomal inhibitors, but not by BFA (S3E Fig). Treatment of 293T-ACE2 cells with these 9 molecules did not reduce cell viability (S3F Fig). These results highlight the significance of lysosomes, but not the Golgi, in SARS-CoV-2 cell entry. Taken together, our results suggest that lysosomes play important roles in viral entry, while the Golgi is important for viral infection at later steps.

### An intact ER-to-Golgi trafficking pathway is essential for SARS-CoV-2 assembly and secretion

To better investigate the role of ER-to-Golgi trafficking in the SARS-CoV-2 viral cycle, we performed a multiple time-point BFA treatment assay (Fig 2C). Huh7-ACE2 cells were infected with SARS-CoV-2 at MOI = 5 for 4 h before treatment with BFA for 4, 6, and 8 h. Both the culture media and cell lysates were collected at the same time for analysis. The 4-hour post infection (hpi) time point was selected based on the observations that strong gRNA signals are detectable at 4 hpi but not 3 hpi, indicating viral entry and active genome replication by this stage [35]. Under these conditions, no significant cell death was observed (Fig 2D).

We found that SARS-CoV-2 secretion was significantly inhibited starting from 4 h BFA treatment, as evidenced by a sharp reduction in N protein levels in the medium (Fig 2E). Importantly, this inhibition of secretion was not due to a block in viral genome replication (Fig 2F). BFA treatment also reduced the intracellular levels of both N and spike proteins, although spike protein appeared less affected (S3G-J Fig). This observation is consistent with the notion that BFA, a known inhibitor of ER-to-Golgi trafficking, not only blocks protein secretion but also partially impairs protein synthesis [36], which may contribute to the observed intracellular reduction.

To more directly assess whether secretion was selectively impaired, we quantified the ratio of viral N protein in the medium versus the cell lysate (M/L ratio) following 6 h of BFA treatment. The M/L ratio of N protein was markedly reduced under BFA treatment (S3K Fig), further supporting the interpretation that secretion is disproportionately affected compared to synthesis or stability. This finding aligns with the view that GRASP55- and TGN46-mediated trafficking events regulate viral protein secretion and that perturbation of this pathway disrupts the efficient release of viral components.

Furthermore, to determine whether SARS-CoV-2 assembly was affected, we measured both the intracellular and released viral titers under the same conditions. The results showed that both the intracellular and extracellular viral titer was significantly decreased by BFA treatment (Fig 2G-H), indicating that an intact ER-to-Golgi trafficking pathway is essential not only for secretion but also for viral assembly.

## SARS-CoV-2 infection alters the Golgi structure

To better understand the role of the Golgi in SARS-CoV-2 infection, we systematically analyzed the Golgi structure in SARS-CoV-2 infected cells by immunofluorescence (IF) and electron microscopy (EM). Using GM130, β-1, 4-Galactosyltransferase 1 (GalT), and Golgin-245 to represent the cis-Golgi, trans-Golgi, and trans-Golgi network, respectively, we found that all sub-compartments of the Golgi were dramatically fragmented after SARS-CoV-2 infection (Fig 3A, F and K). Despite the dispersal of the Golgi in virus-infected cells (Fig 3B-C, G-H, L-M), the Golgi area of all three proteins was not altered. While the expression levels of GM130 and GalT remained unchanged or slightly changed, Golgin-245 level was reduced (Fig 3D-E, I-J, N-O). Both IF and 3D reconstruction of the Golgi by staining of a cis-Golgi marker GM130 demonstrated that spike was highly enriched in the Golgi fragments in SARS-CoV-2 infected cells (Fig 3A, S1 and S2 Videos), indicating an important role of the Golgi in viral infection.

Under EM, the Golgi displayed dramatically different features between uninfected and SARS-CoV-2 infected cells (Figs 3P and S4). In uninfected cells, the Golgi was highly concentrated in the perinuclear region and displayed long cisterna and stacked structures. In infected cells, the Golgi stacks were severely disorganized and fragmented, with most Golgi membranes vesiculated. Interestingly, many virus particles were observed inside the swollen Golgi lumen. Consistent with previous reports [8], we also observed a number of DMVs (Fig 3P, marked by black arrowheads). Taken together, SARS-CoV-2 infection triggers severe Golgi fragmentation.

## SARS-CoV-2 infection down-regulates GRASP55 and up-regulates TGN46 levels

To further investigate the molecular mechanisms of Golgi fragmentation induced by viral infection, we analyzed the morphology and expression level of additional Golgi proteins in SARS-CoV-2 infected cells by immunofluorescence microscopy. Consistent with results from the Golgi markers tested above (Figs 1E, and 3A, F, K), trans-Golgi marker GCC88 and TGN marker Arl1 immunostaining also showed significantly higher fragmentation in infected cells (S5A-C, F-H Fig), with an unchanged total area and marginally changed expression level (S5D-E, I-J Fig). Similarly, the Golgi structure marked by GRASP55 or TGN46 was significantly dispersed with a higher punctum number in infected cells (Fig 4A-C, F–H). However, different from all other Golgi markers tested above, both GRASP55 area and expression level were significantly reduced (Fig 4D-E), whereas both TGN46 area and expression level were significantly increased after SARS-CoV-2 infection (Fig 4I-J).

To further validate the immunofluorescence results, we infected Huh7-ACE2 and Vero E6 cells with SARS-CoV-2 and immunoblotted for key Golgi structural and functional proteins as well as proteins in endosomes, lysosomes, and autophagosomes. Consistent with our immunofluorescence results, GRASP55 down-regulation and TGN46 up-regulation induced by SARS-CoV-2 infection were observed in both cell lines (Figs 4K, L, N, and S5K, L, N), while GRASP65 expression was not altered (Figs 4M and S5M). SARS-CoV-2 infection also increased total CatD level while showing no impact on the expression of EEA1, and GPP130 (Figs 4O-Q and S5O-Q). Given that the infection rate was well below 100%, the actual changes are likely stronger than observed on the western blots. Further, in lung adenocarcinoma Calu-3 cells, SARS-CoV-2 infection also disrupted the Golgi structure and altered GRASP55 and TGN46 protein levels (S5R-W Fig). Taken together, GRASP55 was down-regulated and TGN46 was up-regulated by SARS-CoV-2 infection, suggesting that GRASP55 and TGN46 could serve as potential targets through which SARS-CoV-2 modulates the Golgi structure and function.

## GRASP55 expression significantly reduces SARS-CoV-2 infectivity and secretion

To test whether GRASP55 is a target of SARS-CoV-2 infection, we prepared stable Huh7-ACE2 cells expressing GFP or GFP-tagged wild-type (WT) GRASP55 by lentiviral transduction. Interestingly, GRASP55-GFP expressing cells displayed a significantly lower percentage of viral infection compared to GFP-expressing cells (Fig 5A-B). No significant cell death

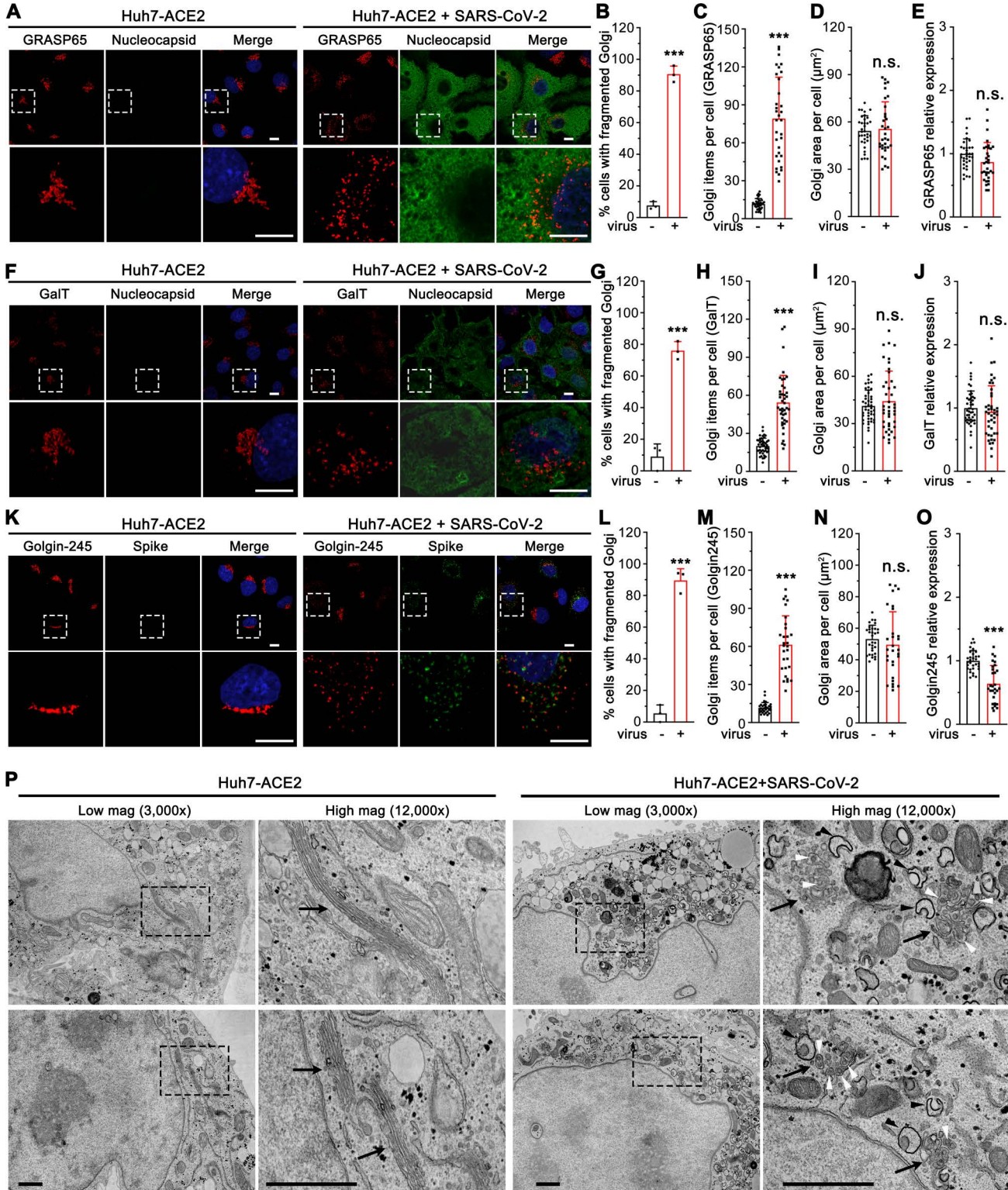

**Fig 3. SARS-CoV-2 infection dramatically disrupts the Golgi structure.** (A) Representative confocal images of Huh7-ACE2 cells incubated with or without SARS-CoV-2 (MOI = 1) for 24 h and stained for a cis-Golgi marker GRASP65 and nucleocapsid. (B-E) Quantification of A for the percentage of cells with fragmented Golgi (B), GRASP65 punctum number (C), area (D), and relative expression level (E). (F) Representative confocal images of

Huh7-ACE2 cells incubated with or without SARS-CoV-2 (MOI = 1) for 24 h and stained for a trans-Golgi marker GalT and nucleocapsid. (G-J) Quantification of GalT in F. (K) Representative confocal images of Huh7-ACE2 cells incubated with or without SARS-CoV-2 (MOI = 1) for 24 h and stained for a TGN marker Golgin-245 and spike. (L-O) Quantification of Golgin-245 in K. Boxed areas in the upper panels of A, F and K are enlarged and shown underneath. Scale bars in all fluorescence images, 10 μm. (P) Electron micrographs of Huh7-ACE2 incubated with or without SARS-CoV-2 (MOI = 1) for 24 h under two different magnifications. Boxed areas on the left images are enlarged on the right. Black arrows, white arrowheads, and black arrowheads indicate Golgi membranes, viral particles, and DMVs, respectively. Scale bars, 500 nm. Data are shown as mean ± SD from three independent experiments. Statistical analyses are performed using two-tailed Student's t-test. *, p < 0.05; ***, p < 0.001; n.s., not significant.

was observed in both cell lines under this condition (Fig 5C). The RNA levels of spike, N, and RdRp showed no significant difference between SARS-CoV-2 infected GFP-expressing and GRASP55-GFP- expressing cells at 4 hpi (Fig 5D), suggesting that viral entry and genome replication process were not unaffected. Consistent with this, spike pseudovirus entry assay also showed that GRASP55 expression does not affect viral entry (S6A-B Fig). To investigate the role of GRASP55 in SARS-CoV-2 trafficking, both cell lysates and culture media from infected GFP- or GRASP55-expressing cells were analyzed by immunoblotting. Both spike and N protein levels were decreased by about the same fraction in the lysate of GRASP55-expressing cells compared to those in GFP-expressing cells (Fig 5E, left panel, F-H), which is consistent with the decrease in viral infection demonstrated by IF (Fig 5A-B). While the spike and N protein levels were significantly reduced in the medium of GRASP55-expressing cells, the spike/N ratio was also significantly lower than that from GFP-expressing cells (Fig 5E, right panel, I-K), suggesting that GRASP55 expression likely inhibits spike incorporation into virions. Further, both TCID50 and plaque assay showed that GRASP55 expression reduced extracellular viral titer (Fig 5L-M).

Previously we have shown that disruption of the stacked Golgi structure by depletion of GRASP55 and GRASP65 accelerates intra-Golgi trafficking, possibly by increasing the membrane surface for vesicle formation [37]. This suggests that SARS-CoV-2 may utilize a similar mechanism to facilitate viral trafficking and release. To test whether GRASP55 expression affects spike protein trafficking, we co-expressed GRASP55 or GFP with spike and performed a cell surface biotinylation assay to compare the amount of spike protein at the plasma membrane. Much less spike was detected at the cell surface in GRASP55-expressing cells compared to GFP-expressing cells (S6C Fig, lane 8 vs. 7), although the total spike expression levels were comparable. This strongly suggests that GRASP55 expression hinders spike protein trafficking. GRASP55 expression also reduced transferrin receptor (TfR) protein level at the cell surface, and this effect was enhanced by spike expression (S6C Fig). Furthermore, Golgi fragmentation induced by SARS-CoV-2 infection was at least partially rescued by GRASP55 expression at all three infection time points (S6D Fig). In addition, GRASP55 expression increased spike-Golgi colocalization in infected cells (S6E-F Fig). Thus, these data suggest that GRASP55 down-regulation contributes to Golgi fragmentation, which is consistent with previous studies, and that other factors may also play a role in Golgi fragmentation in infected cells.

Based on these results, we propose a working model to explain the novel role of GRASP55 in SARS-CoV-2 trafficking (Fig 5N). Under normal conditions, GRASP55 helps to maintain the stacked structure of the Golgi apparatus. SARS-CoV-2 infection decreases GRASP55 expression, resulting in Golgi fragmentation that may in turn facilitate spike trafficking and virion assembly. When GRASP55 is exogenously expressed, spike protein trafficking and virion assembly are reduced, thus limiting infectious virus secretion.

## GRASP55 depletion significantly enhances SARS-CoV-2 infectivity and secretion

To ensure that the inhibition of spike incorporation into virions and viral secretion caused by GRASP55 expression was not a result of global disruption of cell activities, we performed GRASP55 depletion assays. Both spike and N protein levels were significantly increased, with a slight increase in the spike/N ratio in the lysate of GRASP55-depleted cells compared to that in siControl transfected cells (Fig 6A, left panel, B-D). Interestingly, the spike and N protein levels from the medium

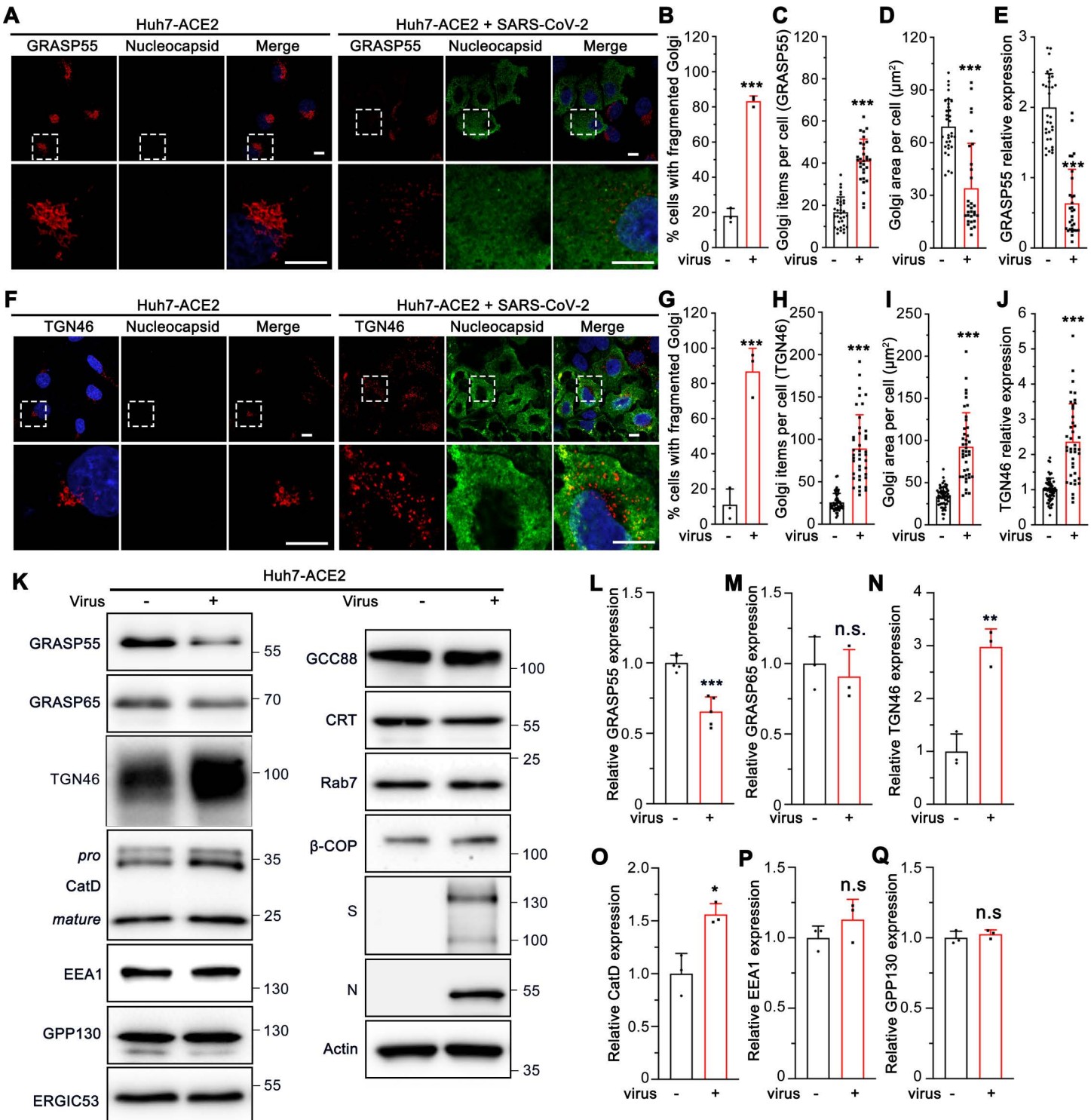

**Fig 4. SARS-CoV-2 down-regulates GRASP55 and up-regulates TGN46 expression in Huh7-ACE2 cells.** (A) SARS-CoV-2 infection reduces GRASP55 expression. Representative confocal images of Huh7-ACE2 cells incubated with or without SARS-CoV-2 (MOI = 1) for 24 h and stained for GRASP55 and nucleocapsid. (B-E) Quantification of the percentage of cells with fragmented Golgi (B), GRASP55 item number (C), area (D), and relative expression level (E) in A. (F) SARS-CoV-2 infection increases TGN46 expression. Representative confocal images of Huh7-ACE2 cells incubated with or without SARS-CoV-2 (MOI = 1) for 24 h and stained for TGN46 and nucleocapsid. (G-J) Quantification of TGN46 in F. Boxed areas in the upper

panels of A and F are enlarged and shown underneath. Scale bars, 10 μm. (K) Immunoblots of Huh7-ACE2 cells incubated with or without SARS-CoV-2 (MOI = 2) for 24 h for indicated proteins. (L-Q) Quantification of the relative expression of GRASP55 (L), GRASP65 (M), TGN46 (N), CatD (O), EEA1 (P), and GPP130 (Q). Data are shown as mean ± SD from at least three independent experiments. Statistical analyses are performed using two-tailed Student's t-test. *, p < 0.05; **, p < 0.01; ***, p < 0.001; n.s., not significant.

of GRASP55-depleted cells were increased by 2.4-fold and 1.6-fold, respectively, and the spike/N ratio was about 1.5-fold higher than that from siControl cells (Fig 6A, right panel, E-G), suggesting that GRASP55 depletion facilitates virion secretion and may alter spike incorporation into virions. To investigate the impact of GRASP55 depletion on viral infection over time, we conducted a viral infection assay at three infection time points, ranging from 5 h to 24 h post-infection (about one to three viral replication cycles). There was no significant difference in the percentage of viral infection between control and GRASP55-depleted cells at 5 h and 8 h post-infection (S7A-B Fig). However, at 24 h, the percentage of viral infection was about 1.5-fold higher in GRASP55-depleted cells compared to control cells (S7A-B Fig), which is consistent with the immunoblotting results (Fig 6A). GRASP55 depletion increased both ACE2 and TfR at the cell surface and disrupted the Golgi structure (S7C-D Fig), supporting that GRASP55 depletion induces Golgi fragmentation and accelerates protein trafficking through the secretory pathway.

To test whether the viral infectious particle production was increased by GRASP55 depletion, we performed TCID50 and plaque assays, showing that GRASP55 depletion significantly increased extracellular viral titer in Huh7-ACE2 cells at 24 hpi (Fig 6H-I). It has been reported that a SARS-CoV-2 infection cycle is 8 hours [38]. We compared both the intracellular viral titer and released viral titer from control and GRASP55-depleted cells within one viral secretion cycle (10 hpi). Consistent with our hypothesis, GRASP55 depletion significantly increased both viral titer inside the cells and in the extracellular space, without causing significant cell death (Fig 6J-L). Taken together, GRASP55 depletion significantly promotes SARS-CoV-2 assembly and secretion.

## TGN46 is required for viral trafficking of both WA-1 and Delta variants of SARS-CoV-2

The finding that TGN46 protein level is significantly increased in SARS-CoV-2 infected cells indicates a possible role for TGN46 in viral infection. Therefore, we speculated that depletion of TGN46 might inhibit SARS-CoV-2 infection. Indeed, TGN46 depletion significantly reduced infection by the SARS-CoV-2 USA-WA1/2020 strain (Fig 7A-B). To determine if this effect is specific for this strain, we also tested the Delta variant. Similar to the WA1/2020 strain, TGN46 depletion also reduced infection by the Delta variant (Fig 7C-D). These results demonstrate a role for TGN46 in SARS-CoV-2 infection independent of the viral strain. Consistent with the viral infection assays, depletion of TGN46 reduced the expression of both spike and nucleocapsid viral proteins when cells were infected with either the USA-WA1 strain or the Delta variant, confirming a critical role for TGN46 in SARS-CoV-2 infection with both strains (Fig 7E). TGN46 depletion had no effect on viral entry (S8A-B Fig), suggesting that TGN46 depletion inhibits a downstream step.

Given that TGN46 is recycling between the Golgi and plasma membrane, we speculated that it may facilitate virion trafficking from the Golgi to the plasma membrane. To test the role of TGN46 in spike trafficking, we transfected control and TGN46-depleted Huh7-ACE2 cells with spike and performed cell surface biotinylation and streptavidin pulldown. The amount of spike protein at the cell surface was significantly reduced by TGN46 depletion (S8C Fig, lane 8 vs. 7). In spike-expressing cells, not only was spike affected by TGN46 knockdown, but several other cell surface proteins, including ACE2, insulin-like growth factor 2 receptor (IGF2R), TfR, and E-cadherin, also displayed a reduction at the plasma membrane in TGN46 knockdown cells (S8C Fig, lane 8 vs. 7). TGN46 knockdown had no effect on ACE2, IGF2R and TfR cell surface localization in non-spike expressing cells (S8C Fig, lane 6 vs. 5), implying that spike expression or viral infection may hijack the majority of TGN46 for its own trafficking, thus impeding the post-Golgi trafficking of many cellular proteins to the plasma membrane. Notably, the level of TGN46 at the cell surface was increased after spike expression

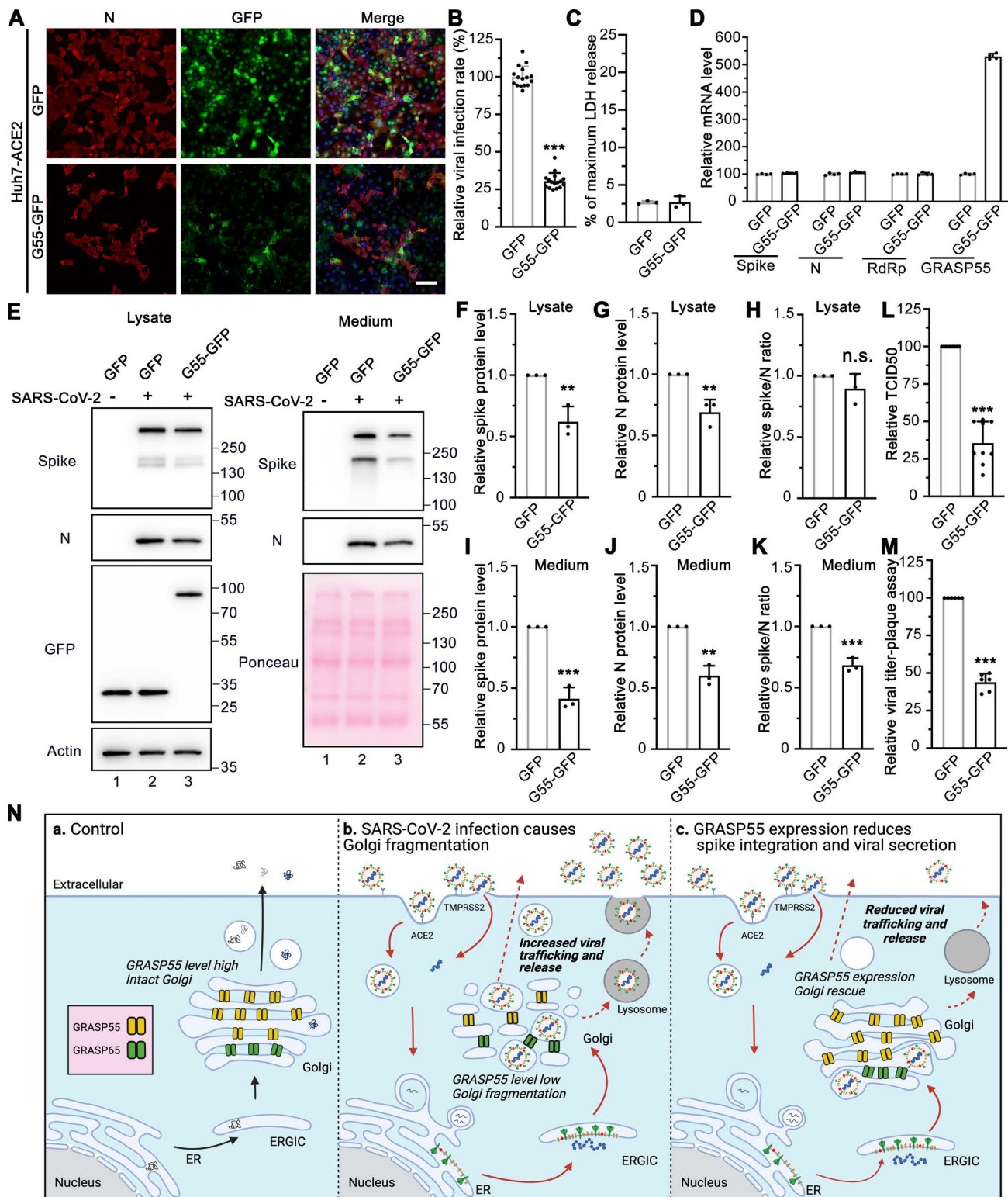

**Fig 5. Expression of GRASP55 reduces SARS-CoV-2 assembly and secretion.** (A) GRASP55 expression reduces SARS-CoV-2 infection. Huh7-ACE2 cells stably expressing GFP or GRASP55-GFP were infected with SARS-CoV-2 (MOI = 3) for 24 h and stained for nucleocapsid. Shown are representative fluorescence images from 15 random images of one representative replicate from three independent experiments. Scale bars, 100 μm.

(B) Quantification of the viral infection percentage in A. (C) Representative LDH assay of Huh7-ACE2 cells stably expressing GFP or GRASP55-GFP infected with SARS-CoV-2 (MOI = 3) for 24 h with 4 technical replicates from two independent experiments. (D) Representative RT-qPCR assay of Huh7-ACE2 cells stably expressing GFP or GRASP55-GFP infected with SARS-CoV-2 (MOI = 5) for 4 h with 4 technical replicates from two independent experiments. (E) Immunoblots of cell lysates and PEG-precipitated culture media of stable Huh7-ACE2 cells expressing GFP or GRASP55-GFP infected with SARS-CoV-2 (MOI = 3) for 24 h for indicated proteins. (F-K) Quantification of the relative expression of spike (F, I), N (G, J), and relative spike/N ratio (H, K) from cell lysates and culture media, respectively. (L-M) TCID50 assay (L) and plaque assay (M) of viruses collected from stable Huh7-ACE2 cells expressing GFP or GRASP55-GFP after SARS-CoV-2 infection (MOI = 3) for 24 h. (N) Proposed working model for a role of GRASP55 in SARS-CoV-2 infection. In brief, under normal conditions (a) GRASP55 is expressed at a high level and maintains the Golgi in an intact structure. After SARS-CoV-2 infection (b), GRASP55 level is reduced, resulting in Golgi fragmentation that may facilitate viral assembly and trafficking. Conversely, when GRASP55 is exogenously expressed (c), the Golgi structure is reinforced and the viral assembly and trafficking speed is limited, thus inhibiting spike incorporation and viral release. Created with BioRender. Data are shown as mean ± SD. Statistical analyses are performed using two-tailed Student's t-test. **, p < 0.01; ***, p < 0.001; n.s., not significant.

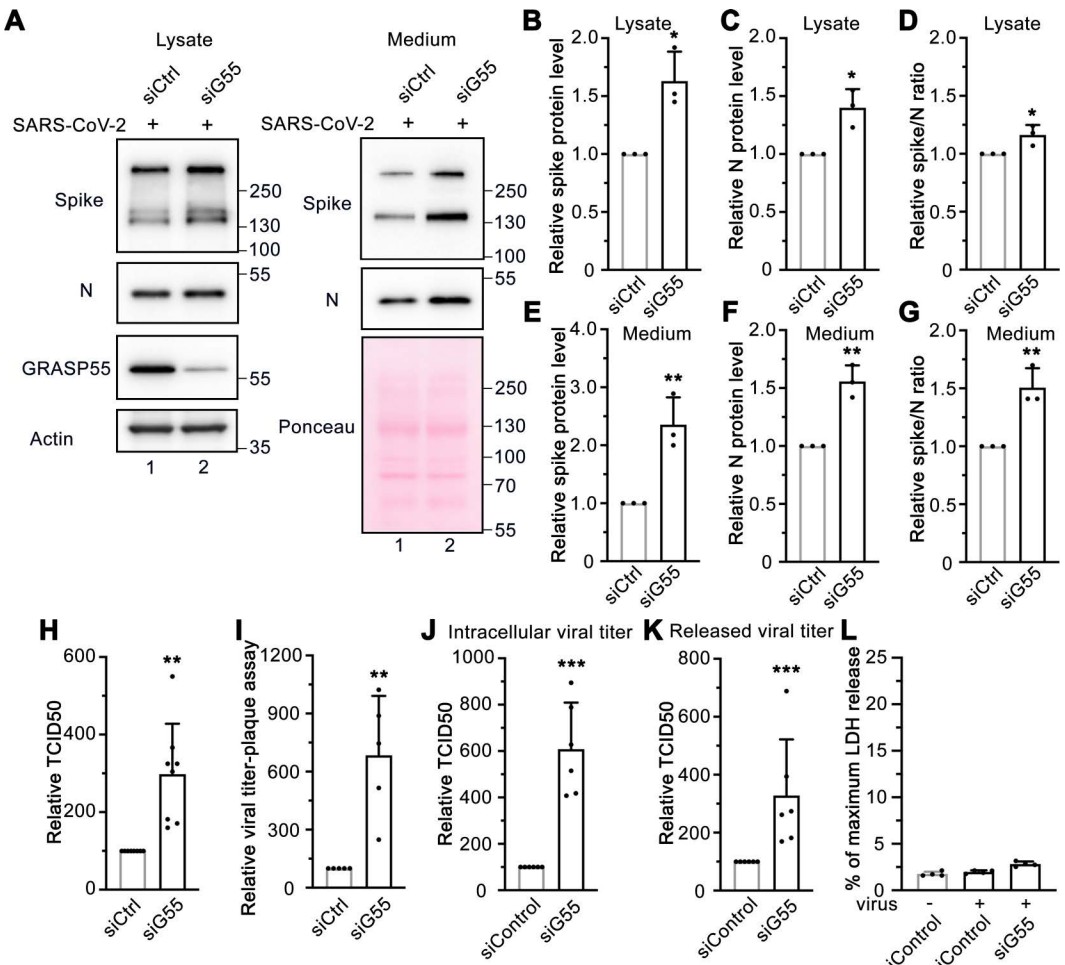

**Fig 6. Depletion of GRASP55 promotes SARS-CoV-2 assembly and secretion.** (A) Immunoblots of cell lysates and PEG-precipitated culture media of Huh7-ACE2 transfected with siControl or siGRASP55 oligos for 48 h followed by SARS-CoV-2 infection (MOI = 3) for 24 h for indicated proteins. (B-G) Quantification of the relative expression of spike (B, E), N (C, F), and relative spike/N ratio (D, G) from cell lysates and culture media, respectively. (H-I) TCID50 assay (H) and plaque assay (I) of viruses collected from Huh7-ACE2 transfected with siControl or siGRASP55 oligos for 48 h followed by infection with SARS-CoV-2 (MOI = 3) for 24 h. (J-K) Intracellular viral titer assay (J) and released viral titer assay (K) of Huh7-ACE2 cells transfected with siControl or siGRASP55 oligoes and infected with SARS-CoV-2 (MOI = 5) for 10 h with 6 replicates from two independent experiments. (L) Representative LDH assay of Huh7-ACE2 cells transfected with siControl or siGRASP55 oligoes and infected with SARS-CoV-2 (MOI = 5) for 10 h with 4 technical replicates from two independent experiments. Data are shown as mean ± SD. Statistical analyses are performed using two-tailed Student's t-test. *, p < 0.05; **, p < 0.01; ***, p < 0.001; n.s., not significant.

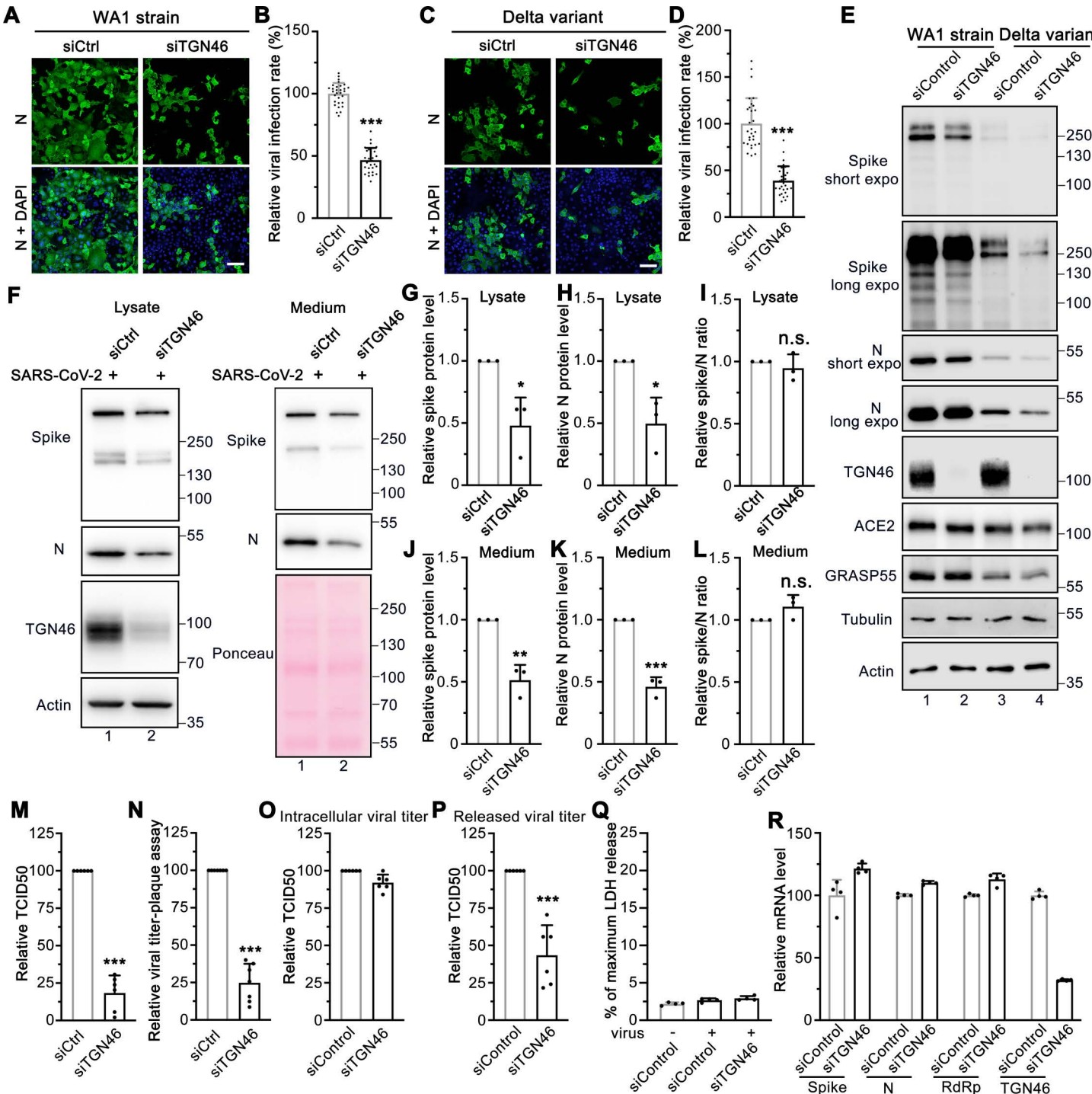

**Fig 7. TGN46 depletion impairs SARS-CoV-2 trafficking.** (A, C) TGN46 depletion reduces SARS-CoV-2 infection. Huh7-ACE2 cells were transfected with siControl or siTGN46 oligos for 48 h followed by infection (MOI = 1) with the WA1 strain (A) or Delta variant (C) of SARS-CoV-2 for 24 h and stained for nucleocapsid. Shown are representative fluorescence images from 30 random images from two independent experiments. Scale bars, 100 µm. (B, D) Quantification of the viral infection percentage in A and C, respectively. (E) TGN46 depletion reduces viral protein expression in host cells. Huh7-ACE2 cells were transfected with siControl or siTGN46 oligos for 48 h followed by infection with SARS-CoV-2 WA1 strain and Delta variant (MOI = 3) for 24 h. Cell lysates were blotted for indicated proteins. Long and short exposures are shown for spike and N proteins. (F) Immunoblots of cell lysates and PEG-precipitated culture media of Huh7-ACE2 transfected with siControl or siTGN46 oligos for 48 h followed by SARS-CoV-2 infection (MOI = 3) for 24 h

for indicated proteins. (G-L) Quantification of the relative expression of spike (G, J), N (H, K), and relative spike/N ratio (I, L) from cell lysates and culture media, respectively. (M-N) TCID50 assay (M) and plaque assay (N) of viruses collected from Huh7-ACE2 transfected with siControl or siTGN46 oligos for 48 h followed by infection with SARS-CoV-2 for 24 h. (O-P) Intracellular viral titer assay (O) and released viral titer assay (P) of Huh7-ACE2 cells transfected with siControl or siTGN46 oligoes and infected with SARS-CoV-2 (MOI = 5) for 10 h with 6 replicates from two independent experiments. (Q) Representative LDH assay of Huh7-ACE2 cells transfected with siControl or siTGN46 oligoes and infected with SARS-CoV-2 (MOI = 5) for 10 h with 4 technical replicates from two independent experiments. (R) Representative RT-qPCR assay of Huh7-ACE2 cells transfected with siControl or siTGN46 oligoes and infected with SARS-CoV-2 (MOI = 5) for 4 h with 4 technical replicates from two independent experiments. Data are shown as mean ± SD. Statistical analyses were performed using two-tailed Student's t-test. *, $p < 0.05$; **, $p < 0.01$; ***, $p < 0.001$; n.s., not significant.

(S8C Fig, lane 7 vs. 5), indicating that TGN46 may recycle more frequently between the TGN and the plasma membrane when spike is expressed.

To determine if TGN46 regulates spike incorporation into virions, both cell lysate and medium were analyzed by immunoblotting (Fig 7F). Both spike and N were decreased in cell lysate and medium of TGN46-depleted cells (Fig 7G-I, J–L), similar to GRASP55 expression. However, the spike/N ratio remained unchanged in both lysate and medium after TGN46 depletion (Fig 7I, L), suggesting that TGN46, unlike GRASP55, does not modulate spike incorporation into virions. Consistent with this model, TGN46 expression did not disrupt the Golgi structure (S8D Fig).

TGN46 depletion reduced infectious viral production, as shown by both TCID50 and plaque assays (Fig 7M-N). Furthermore, at 10 hpi within one viral secretion cycle, TGN46 depletion significantly reduced viral titer in the extracellular space, but not inside the cells (Fig 7O-P), suggesting that TGN46 regulates SARS-CoV-2 secretion, but not assembly. Cell death was not induced under this condition (Fig 7Q). TGN46 depletion did not inhibits SARS-CoV-2 entry and genome replication at 4 hpi (Fig 7R). Taken together, we propose a working model of TGN46 in SARS-CoV-2 trafficking (S8E Fig). In brief, SARS-CoV-2 infection significantly increases TGN46 protein level to accelerate virion secretion without affecting viral assembly and entry.

## Discussion

In this study, we demonstrated that SARS-CoV-2 infection triggers a global change of the endomembrane system of host cells, particularly affecting the Golgi structure. Conversely, Golgi disruption by BFA strongly inhibits viral assembly and secretion but not viral entry. To determine the mechanism of Golgi fragmentation induced by SARS-CoV-2 infection, we surveyed a large number of Golgi proteins in SARS-CoV-2 infected cells. While several Golgi proteins are impacted by viral infection, GRASP55 and TGN46 are the top two proteins whose levels change most dramatically in opposite trends. Surprisingly, overexpression of GRASP55 inhibits virion assembly, while GRASP55 depletion reverses the effects. Distinct from GRASP55, TGN46 plays a role in SARS-CoV-2 secretion but not assembly, suggesting that GRASP55 and TGN46 modulate SARS-CoV-2 replication at different stages. This is consistent with their localizations that GRASP55 is localized at medial/trans Golgi, while TGN46 recycles between the TGN and the plasma membrane. Manipulation of either GRASP55 or TGN46 does not affect viral entry. Thus, our study uncovers a novel mechanism by which SARS-CoV-2 modulates Golgi structure and function via regulating GRASP55 and TGN46 expression to facilitate viral assembly and secretion. Our results suggest that the Golgi apparatus may serve as a novel therapeutic target for the treatment of COVID-19 and other diseases caused by viruses that utilize a similar trafficking pathway.

The Golgi stacking proteins, GRASP55 and GRASP65, play essential roles in Golgi structure formation [39,40] by forming trans-oligomers that "glue" adjacent Golgi cisternae together into stacks and ribbon [41,42]. Expression of phospho-deficient mutants of GRASPs (e.g., the GRASP domain of GRASP55 or GRASP55) at least partially inhibits mitotic Golgi disassembly [39,43]. Inhibition of Golgi stacking in cells by microinjecting GRASP antibodies or by GRASP55/65 knockdown/knockout [41,44] accelerates conventional protein trafficking but impairs accurate glycosylation and sorting [37,45,46], increases heparan sulfate but decreases chondroitin sulfate synthesis [47], and reduces unconventional protein secretion of mutant huntingtin [48]. A plausible explanation is that stacking may reduce the accessibility of coat

proteins to Golgi membranes, which decreases the rate of vesicle budding and fusion [49–51]. Therefore, it is reasonable to speculate that Golgi fragmentation observed in this study may facilitate SARS-CoV-2 assembly and release.

SARS-CoV-2 infection down-regulates GRASP55 but not its homolog GRASP65. GRASP55 has been shown to play a crucial role in various stress responses [52–55], while GRASP65 functions more in cell migration and apoptosis [56–58]. In this study, we found that GRASP55 plays a vital role in viral assembly and secretion, likely due to its role in Golgi structure formation. Indeed, GRASP55 expression significantly inhibited spike protein trafficking to the cell surface, supporting our hypothesis that SARS-CoV-2 induced GRASP55 down-regulation facilitates viral assembly and spike incorporation into virions, consistent with the previous finding that GRASP55-depletion induced Golgi structural defect enhances conventional protein trafficking [37]. It is believed that SARS-CoV-2 virions assemble within the ERGIC [22] and Golgi [8,17]. Therefore, it is reasonable that GRASP55 regulates SARS-CoV-2 spike incorporation into virions. Given the pivotal role of the spike protein in viral infection, the finding that GRASP55, a Golgi structural protein, regulates both viral production and infectivity is intriguing. It would be compelling to determine whether M or E protein is also increased on each virion by GRASP55 depletion.

Many enveloped and non-enveloped viruses display trimeric receptor-binding proteins at the viral surface (e.g., SARS-CoV-2 spike, influenza HA) that are usually about 10 nm or longer to play a crucial role in mediating cell entry [59]. The number of these receptor-binding proteins per virion is a determinant of viral infectivity. Among enveloped RNA viruses, this number varies from 7-14 (HIV), 60 (Zika virus), 400–500 (influenza virus), to 1200 (vesicular stomatitis virus). Three independent teams reported that there were on average 24–40 spikes on each SARS-CoV-2 virion [60–62]. In our study, we found that GRASP55 depletion increased the ratio of extracellular spike/N protein, suggesting that it may increase the spike number on secreted SARS-CoV-2 virions (Fig 6A-G). Whether GRASP55 depletion affects SARS-CoV-2 viral particle size remains an open and interesting question.

TGN46 is a glycoprotein that recycles between the TGN and cell surface [63,64]. TGN46 was previously reported to directly interact with integrin β1 to modulate its trafficking [65], while recently TGN46 was reported to serve as a sorting receptor at the TGN for CARTS, a class of protein kinase D-dependent TGN-to-plasma membrane carriers [66]. Therefore, we postulate that the increased TGN46 level indicates a need for TGN46 to accommodate the high flux of virion trafficking through the late secretory pathway, and our results support this hypothesis. Notably, depletion of TGN46 inhibited spike protein trafficking, but not the amount of ACE2 at the cell surface, while spike expression increased the TGN46 level at the cell surface. These observations indicate that TGN46 functions in the post-Golgi trafficking of the SARS-CoV-2 virus. Our discoveries that SARS-CoV-2 reduces GRASP55 and elevates TGN46 expression are also supported by a number of RNA-seq and proteomic studies of SARS-CoV-2 infected cells and human tissues [67–69]. Significantly, our results demonstrate that either overexpression of GRASP55 or depletion of TGN46 inhibits SARS-CoV-2 secretion. To be noted, the mechanisms underlying SARS-CoV-2-induced GRASP55 down-regulation and TGN46 up-regulation - whether through altered gene expression or protein stability - remain unclear and warrant further investigation.

It is noteworthy that both Tg and BFA inhibit viral infection but promote viral entry. It has been reported that short-term BFA treatment (4–6 h) did not affect mouse hepatitis virus (MHV) egress [12]. In contrast, our 4 h BFA treatment significantly inhibited SARS-CoV-2 assembly and secretion without inhibiting viral cell entry and genome replication. This suggests that the integrity of the conventional trafficking pathway is essential for SARS-CoV-2 trafficking and spread and it is likely that SARS-CoV-2 and MHV employ different pathways for viral secretion. Therefore, our study demonstrated, for the first time, that short-term BFA treatment within one virion cycle inhibits SARS-CoV-2 assembly and secretion, highlighting the dependence of SARS-CoV-2 assembly and secretion on the ER-to-Golgi trafficking route. It would be interesting to investigate whether BFA treatment similarly inhibits the assembly and trafficking of other coronaviruses. Although CQ has been reported not to block SARS-CoV-2 infection in the human lung cell line Calu-3 [70], it greatly reduced both the percentage of infected cells and the efficiency of viral entry in Huh7-ACE2 cells in our study. This discrepancy may be explained by the fact that SARS-CoV-2 entry into Calu-3 cells is independent of endosomal acidification due to its high

expression level of TMPRSS2, which activates spike at the plasma membrane; while Huh7.5 (a derivative cell line of Huh7) cells heavily rely on the endosomal acidity for SARS-CoV-2 entry [71], leading to a high dependence on lysosomes whose function can be inhibited by CQ. Vacuolin-1 is sometimes regarded as a lysosomal exocytosis inhibitor, but it has also been reported to alter lysosome morphology without inhibiting Ca2+-regulated lysosomal exocytosis [72]. Although we showed that vacuolin-1 inhibits viral entry, it is unclear whether the inhibition of SARS-CoV-2 infection by vacuolin-1 observed in our study is also contributed by the inhibition of SARS-CoV-2 release via lysosomal exocytosis.

Currently, it is still debated about whether SARS-CoV-2 virions transit the Golgi during secretion [23]. However, our study strongly supports that Golgi is required for SARS-CoV-2 secretion for the following reasons. Firstly, we observed that many mature SARS-CoV-2 virions reside in both rims of a stacked Golgi and fragmented Golgi remnants by electron microscopy. Secondly, blocking ER-to-Golgi trafficking almost completely inhibits SARS-CoV-2 assembly and secretion. Thirdly, two Golgi proteins, GRASP55, whose expression level is closely related to Golgi structure, and TGN46, which constitutively recycles from TGN and plasma membrane, play important roles in SARS-CoV-2 virion assembly and secretion, suggesting that the Golgi is a key target utilized by SARS-CoV-2. Lastly, our findings that GRASP55 protein may regulate SARS-CoV-2 spike incorporation into virions support that the virion assembly occurs in the secretory pathway. In summary, our study reveals that the Golgi plays a key role in SARS-CoV-2 assembly and secretion, and it could be a potential target for blocking the spread of different SARS-CoV-2 variants.

## Methods and materials

### Ethics statement

All virus inactivation protocols were validated and approved (IBCA00000123) by the University of Michigan Institutional Biosafety Committee, Subcommittee for Work with BSL3 Agents, including select agents and toxins.

### Generation of cDNA constructs

All pCAG plasmids encoding Strep-tagged SARS-CoV-2 proteins were kind gifts from Dr. Nevan Krogan [5]. pHAGE2-pEF1α-GRASP55-GFP was constructed by cloning GRASP55-GFP from pEGFP-N1-GRASP55 into the pHAGE2- pEF1α vector. psPAX2 and pMD2G were obtained from Addgene. A C-terminal 19 aa truncated SARS-CoV-2 spike expression vector and CMV-eGF1 were kind gifts from Dr. Marilia Cascalho.

### Cell culture and transfection

ACE2-expressing Huh7 cells were sorted, enriched, and selected by flow cytometry (FACS) to optimize SARS-CoV-2 infection efficiency [29]. Huh7-ACE2 and Vero E6 (ATCC CRL 1586) cells were cultured in Dulbecco's Modified Eagle's Medium (Gibco) supplemented with 10% fetal bovine serum (FBS, Hyclone), 100 units/ml penicillin and 100 µg/ml streptomycin (Invitrogen) at 37°C with 5% $CO_2$. Plasmid transfection was performed using Lipofectamine 2000 or Lipofectamine 3000 (Invitrogen) following the manufacturer's instructions. After 24–48 h, transfected cells were fixed by 4% paraformaldehyde (PFA) for immunofluorescence analysis or were lysed in IGEPAL-C360 lysis buffer (20 mM Tris HCl pH 8.1, 37 mM NaCl, 1% IGEPAL-C360, 10% glycerol, 2 mM EDTA supplemented with protease inhibitor cocktail from ThermoFisher) for immunoblotting.

### SARS-CoV-2 viral strains and amplification

Most experiments were conducted using SARS-CoV-2, Isolate USA-WA1/2020 (BEI NR-52281), unless specified. USA-WA1/2020 was isolated from an oropharyngeal swab from a patient who returned from an affected region of China and developed COVID-19 on January 19, 2020, in Washington, USA. The Delta variant of SARS-CoV-2 used in the study is the Isolate hCoV-19/USA/PHC658/2021 (Lineage B.1.617.2; BEI NR-55611). SARS-CoV-2 working stocks were

amplified by infecting Vero E6 cells in 2% FBS DMEM and 182 cm$^2$ flasks with 70–80% confluency. Flasks were incubated until cytopathic effect (CPE) became distinctly visible, generally at 2–4 days post-infection (dpi). The cell debris was pelleted and removed by centrifugation, and the supernatant was harvested, filtered through 0.45 μm SCFA syringe filters and pipetted into 800 μl aliquots, with tubes of viral stock then being stored at -80ºC. One vial of each propagated stock was thawed after 24 h and titer was measured by TCID50, with positive wells determined by the presence of CPE 4 dpi observed by 4x objective phase-contrast microscopy. Viral titer was calculated by the modified Reed and Muench method.

## RNA interference

siRNA Universal Negative Control #1 (siControl), siGRASP55 with 5'-UGAUAACUGUCGAGAAGUGAUUAUU-3' sequence, siGRASP65 with 5'-CCUGAAGGCACUACUGAAAGCCAAU-3' sequence, and siTGN46 with 5'- CCAC CGAAAGCGUCAAGCAAGAAGA-3' sequence were obtained from Sigma-Aldrich. Huh7-ACE2 cells were transfected with siRNA oligos (final concentration, 50 nM) by RNAi-MAX (Invitrogen) on day 1 and infected with SARS-CoV-2 on day 3. Cells were fixed by 4% paraformaldehyde (PFA) for 30 min on day 4 for immunofluorescence analysis.

## Immunoblotting

Huh7-ACE2 or Vero E6 cells were seeded onto 6-well plates on day 1 and incubated with or without SARS-CoV-2 (MOI = 2) on day 2 for 24 h. Cells were lysed in IGEPAL-C360 lysis buffer. For regular immunoblotting of uninfected cells, cells were lysed in lysis buffer (20 mM Tris-HCl pH 8.0, 150 mM NaCl, 1% Triton X-100 supplemented with protease inhibitor cocktail (Thermo Fisher). The homogenate was centrifuged at 13000 × $g$ for 15 min to remove cell debris, then denatured at 95°C for 10 min in 2x Laemmli buffer supplemented with 5% 2-mercaptoethanol. Protein quantification was performed using a Bradford Kit (Bio-Rad). Protein samples were analyzed by SDS-PAGE and then transferred to nitro-cellulose membranes using a semi-dry or wet transfer machine. The membranes were blocked in 5% milk in PBST (0.1% Tween 20 in phosphate buffered saline) and incubated with proper antibodies and visualized by a FluorChem M chemi-luminescent imager (ProteinSimple, San Jose, CA) with enhanced chemiluminescence (Thermo Fisher). The antibodies used in this study are shown in Supplementary Table 1.

## Immunofluorescence

Huh7-ACE2 or Vero E6 cells were seeded on poly-lysine-coated coverslips. SARS-CoV-2 infected cells were fixed with 4% PFA for 30 min for complete virus inactivation. Cells were quenched with 50 mM NH$_4$Cl in PBS for 10 min with gentle rocking and permeabilized with 0.2% Triton X-100 for 10 min. Cells were blocked with 0.2% gelatin blocking buffer in PBS (PGA) for 30 min at room temperature, incubated with primary antibodies overnight and secondary antibodies for 1 h. The antibodies used for immunofluorescence were shown on S1 Table. Hoechst 33258 (Sigma-Aldrich) was used to stain the nuclear DNA. About 15–20 random images were taken for each independent experiment with a 60x oil objective on a Nikon ECLIPSE Ti2 Confocal microscope and processed with maximum intensity projection. Quantifications were performed to calculate item number, area, and sum intensity of selected ROIs using the Nikon NIS-Elements AR analysis software. Control and virus-infected samples were processed in parallel following the same procedure. All images for the same marker were captured and processed with the same setting. For 3D video reconstruction, samples were prepared as above. Z-stacks of 40 images were taken in 0.15 μm increments. Maximum intensity projection was performed, and videos were made with the Nikon NIS-Elements AR analysis software.

## Quantification of viral infection

Huh7-ACE2 cells were seeded onto poly-lysine-coated coverslips on day 1. On day 2, cells were treated with or without indicated chemicals and immediately infected with SARS-CoV-2 for 24 h. Cells were fixed with 4% PFA for 30 min and

processed for immunofluorescence analysis. Images were taken with a 20x air objective on a Nikon ECLIPSE Ti2 Confocal microscope and processed with maximum projection. The number of infected and uninfected cells were quantified by the counting tool in Adobe Photoshop using nucleocapsid as a virus marker. The viral infection percentage was calculated as [infected cells/(uninfected + infected cells)], and values for the DMSO treatment group were normalized to 100%.

## TCID50 assay

Vero E6 cells were seeded into a 96-well plate at the seeding density of 1 x $10^4$ cells/well in 100 µl full culture medium. When cells were close to confluency, 125 µl pure DMEM medium without FBS was added to each well. 25 µl viral stock was added to the first column (8 wells) and mixed well. 25 µl mixture from the first column was taken and added into the second column. This step was repeated to obtain a serial dilution with 8 replicates for each sample and 7 dilutions. After incubation for 4–5 days, the cells were observed under microscope for each well and TCID50 was calculated.

## Plaque assay

Vero E6 cells were seeded into a 6-well plate at the seeding density of 3 x $10^5$ cells/well. When cells were close to confluency, culture medium was discarded and 1 ml DMEM medium with 2% FBS was added. 100 µl virus with a total of 5 serial dilution was added. Cells were incubated with viruses for 1 h and mixed by moving plates back and forth every 15 min. Infection medium was aspirated and 2 ml pre-warmed carboxymethylcellulose (CMC) overlay medium (DMEM medium with 2.5% FBS and 1% CMC) was added. Cells were incubated in the incubator for 4–5 days. CMC overlay medium was aspirated, and cells were incubated with 0.5 ml 4% PFA in PBS for 20 min at room temperature. PFA was removed before adding 0.1% crystal violet in 20% methanol for 20 min at room temperature. Cells were gently rinsed with water. Plaques were counted after drying and virus titer was calculated.

## Intracellular viral titer assay

Infected cells in 6-well plates were washed once with chilled PBS to remove extracellular virions. Cells were incubated with chilled PBS and immediately stored at -80ºC overnight. Cells were incubated for 15 minutes at 37˚C to fully lyse the cells and release intracellular viral particles. After centrifugation at 3,200 *g* for 15 minutes, the supernatant was transferred into a new tube and added to Vero E6 cells for titer measurement.

## PEG virus precipitation assay

PEG precipitation was performed to concentrate viruses from the culture medium of Huh7-ACE2 cells, by following the product instructions (Cat #MAK343, Sigma). In brief, after 24 h infection, cells were lysed using RIPA buffer, and the 10 ml media were centrifuged at 3,200 g for 15 min at 4ºC to remove cell debris. Supernatants were transferred into new tubes and mixed with 2.5 ml PEG solution (5x) at 4ºC overnight. After centrifugation at 3,200 g for 30 min, supernatant was carefully removed, and the white pellet was resuspended by 100 µl virus resuspension solution and added with RIPA buffer, then denatured at 95˚C for 10 min in 2x Laemmli buffer supplemented with 5% 2-mercaptoethanol for WB analysis.

## BFA time-point assay

Huh7-ACE2 cells were seeded at a density of 3.5 x $10^5$ cells/ml on day 1. On day 2, cells were infected with SARS-CoV-2 WA-1 at MOI = 5. Viruses were washed out 1 h later, incubated in 2% serum culture medium for another 3 h, and treated with 5 µg/ml BFA for 4, 6, and 8 h. Cells were collected for the LDH assay and RT-qPCR assay, while media were collected for PEG precipitation for western blotting analysis. Cells and media were also collected for the intracellular and released viral titer assay.

### Stable cell lines preparation by lentivirus

HEK293T cells were seeded into 10-cm dishes at a seeding density of 300,000 cells/ml on day 1. On day 2, cells were co-transfected with three plasmids, pHAGE2-pEF1α-GFP or pHAGE2-pEF1α-GRASP55-GFP, psPAX2, and pMD2.G, by Lipofectamine 3000. Culture medium was changed 6 h post transfection. Lentiviruses were collected after 24 h and 48 h transfection, filtered using polyethersulfone (PES) filter with a 0.45 µm pore size, and stored at -80ºC. Huh7-ACE2 or 293T-ACE2 cells were seeded into 6-cm dishes. Next day, cells were transduced by lentiviruses expressing GFP or GRASP55-GFP with 10 µg/ml polybrene. Puromycin at a final concentration of 2 µg/ml was added into cells 48 h post infection. Transduction efficiency was validated by immunofluorescence after continuous puromycin treatment for 3–4 days.

### Spike pseudotyped lentivirus preparation

Spike pseudotyped lentiviruses were prepared using 293T cells co-transfected with pLenti-pseudo spike (a SARS-CoV-2 spike expression vector), psPAX2, and CMV-eFP1 by Lipofectamine 3000. All other steps are the same with GRASP55-GFP expressing lentivirus preparation as described above. Virus titer was measured by using Lenti-X p24 Rapid Titer Kit (Cat #022261, Takara).

### Virus cell entry assay

293T-ACE2 cells were seeded into 6-well plates at the seeding density of 200,000 cells/ml on day 1 and transfected with RNAi oligoes on day 2. On day 3, cells were seeded into 96-well plates at the seeding density of 300,000 cells/ml on day 3. On day 4, cells were infected by Spike pseudotyped lentiviruses with 8 µg/ml polybrene. On day 5, cells were lysed with the Bright Glo Kit reagent for 5 min followed by immediate reading using Tecan SparkCyto with the luminescence mode. For drug treatment, cells were seeded on day 1, and on day 2 cells were infected by Spike pseudotyped lentiviruses and co-treated with small molecules. All experiments were done in four replicates and repeated at least three times.

### Authentic SARS-CoV-2 entry assay

Huh7-ACE2 cells in 6-well plates were placed at 4ºC for 15 minutes before infection with SARS-CoV-2 at MOI = 5. Cells were kept at 4ºC for another hour followed by washing using 2 ml chilled PBS to remove unattached virions. Cells were moved to the 37ºC incubator to allow virus entry into the cells for another 4 h. Cells were washed by chilled PBS once and treated with TRI Reagent for RNA isolation and RT-qPCR analysis.

### RT-qPCR

Cells were washed once with chilled PBS and treated with TRI Reagent. RNA was isolated from cells by following the manufacturer's instructions (Zymo Direct-zol RNA Miniprep Kits, R2053). RNA products were reverse transcribed to cDNA using GoScript Reverse Transcription System (Promega, A5001)). RT–qPCR was performed using SYBR Select Master Mix (Applied Biosystems, A25742) on a QuantStudio 5 Flex machine according to the manufacturer's instructions. The expression levels of the target genes were calculated using $2^{-\Delta\Delta CT}$ and normalized to that of Actin. Primer sequences for N, S, and RdRp were from Origene. All primers used are listed in S2 Table.

### LDH release assay

LDH release assay was performed by following the manufacturer's instructions (J2381, Promega). In brief, cells were treated with small molecules of interests or infected with SARS-CoV-2 for the indicated time points in the Figure legend. Culture media were collected and mixed with LDL storage buffer at a 1:24 ratio. Medium collected from cells treated with

0.2% Triton X-100 for 15 minutes as positive control. 50 µl LDH detection reagent was added to 50 µl medium and LDH storage buffer mixture. Luminescence was measured immediately after 30 minutes of incubation.

## CellTiter-Glo cell viability assay

293T-ACE2 cells were seeded into 96-well plates at a density of 300,000 cells/well on day 1. On day 2, cells were treated with small molecules for 24 h. On day 3, cells were lysed with the CellTiter-Glo reagent (G9242, Promega) for 10 min followed by immediate luminescence reading.

## Cell surface biotinylation assay

Cell surface biotinylation assay was performed as previously described [56]. All procedures were performed on ice or at 4°C. In brief, Huh7-ACE2 cells were seeded on 15-cm dishes. After knocking down TGN46 or overexpression of GRASP55, cells were washed twice with ice-cold PBS, treated with 10 ml of 0.5 mg/ml NHS-SS-biotin (Thermo Fisher) in PBS for 20 min in the cold room, and quenched by 100 mM glycine in PBS for 10 min. After three washes with ice-cold PBS, cells were lysed in lysis buffer (20 mM Tris-HCl, pH 8.0, 150 mM NaCl, 1% Triton X-100, 0.1% SDS, 0.1% sodium deoxycholate, 1 mM EDTA, 50 mM sodium fluoride, 20 mM sodium orthovanadate, and 1×protease inhibitor cocktail). After centrifugation, the supernatants were adjusted to the same concentration and incubated with streptavidin-agarose beads overnight. After extensive wash, beads were boiled in SDS loading buffer with 40 mM DTT. Proteins were separated by SDS-PAGE and analyzed by immunoblotting.

## Electron microscopy (EM)

All EM-related reagents were purchased from Electron Microscopy Sciences (EMS; Hatfield, PA). Huh7-ACE2 cells were infected with or without SARS-CoV-2 (MOI = 1) for 24 h and fixed in 2% glutaraldehyde at room temperature for 30 min. Cells were washed three times with 0.1 M sodium cacodylate, and post-fixed on ice in 1% (weight/vol) reduced osmium tetroxide, 0.1 M sodium cacodylate and 1.5% (weight/vol) cyanoferrate. Cells were rinsed 3 times with 0.1 M sodium cacodylate and 3 times with water, and processed for successive dehydration and embedding as previously described [73]. Resin blocks were cut to ultrathin (50–70 nm) sections with a diamond knife and mounted on Formvar-coated copper grids. Grids were double contrasted with 2% uranyl acetate for 5 min and then with lead citrate for 2 min and washed with excess water. Images were captured at 1,500x, 3,000x, 8,000x, and 12,000x magnifications by a JEOL JEM-1400 transmission electron microscope.

## Quantitation and statistics

All data represent the mean ± SD of at least three independent experiments unless noted. Statistical analyses were performed by two-tailed Student's t-test, or one-way ANOVA with Tukey's multiple comparison test using GraphPad Prism9. Differences in means were considered as no statistical significance (n.s.) if $p \geq 0.05$. Significance levels are as follows: *, $p < 0.05$; **, $p < 0.01$; ***, $p < 0.001$. Figures are assembled with Photoshop CS6 Extended (Adobe, San Jose, CA). Figs 5N and S8E are created with BioRender.

## Supporting information

**S1 Fig. SARS-CoV-2 infection modulates the formation of COPI and COPII vesicles.** (A) Representative confocal images of Huh7-ACE2 cells incubated with or without SARS-CoV-2 (MOI = 1) for 24 h and stained for calreticulin and nucleocapsid. (B-C) Quantification of calreticulin for the area (B) and relative expression (C) in A. (D) Representative confocal images of Huh7-ACE2 cells incubated with or without SARS-CoV-2 (MOI = 1) for 24 h and stained for Sec31 and nucleocapsid. (E-G) Quantification of Sec31 item number (E), area (F), and relative expression (G) in D. (H)

Representative confocal images of Huh7-ACE2 cells incubated with or without SARS-CoV-2 (MOI = 1) for 24 h and stained for β-COP and nucleocapsid. (I-K) Quantification of β-COP in H. (L) Representative confocal images of Huh7-ACE2 cells incubated with or without SARS-CoV-2 (MOI = 1) for 24 h and stained for clathrin and spike. (M-O) Quantification of clathrin in L. Boxed areas in the upper panels are enlarged and shown underneath. Scale bars in all panels, 10 μm. All quantitation data are shown as mean ± SD from three independent experiments. Statistical analyses were performed using two-tailed Student's t-test. *, p < 0.05; ***, p < 0.001; n.s., not significant.
(TIF)

**S2 Fig. SARS-CoV-2 infection increases autophagy but exhibits minor effects on cytoskeleton.** (A) Representative confocal images of Huh7-ACE2 cells incubated with or without SARS-CoV-2 (MOI = 1) for 24 h and stained for LC3 and nucleocapsid. (B-D) Quantification of LC3 item number (B), area (C), and relative expression (D) in A. (E) Representative confocal images of Huh7-ACE2 cells incubated with or without SARS-CoV-2 (MOI = 1) for 24 h and stained for cathepsin D and nucleocapsid. (F-H) Quantification of cathepsin D in E. (I) Representative confocal images of Huh7-ACE2 cells incubated with or without SARS-CoV-2 (MOI = 1) for 24 h and stained for α-Tubulin and spike. (J-K) Quantification of tubulin in I. (L) Representative confocal images of Huh7-ACE2 cells incubated with or without SARS-CoV-2 (MOI = 1) for 24 h and stained for actin (with phalloidin) and spike. (M-N) Quantification of actin in L. Boxed areas in the upper panels are enlarged and shown underneath. Scale bars in all panels, 10 μm. (O) Representative LDH assay of Huh7-ACE2 cells infected with SARS-CoV-2 at different MOIs for 24 h with 4 technical replicates from two independent experiments. All quantitation data are shown as mean ± SD from three independent experiments. Statistical analyses were performed using two-tailed Student's t-test. *, p < 0.05; ***, p < 0.001; n.s., not significant.
(TIF)

**S3 Fig. Molecules disrupting Golgi functions inhibit SARS-CoV-2 infection at low concentrations.** (A) Representative confocal images of Huh7-ACE2 cells infected with SARS-CoV-2 for 24 h in the presence of indicated molecules and stained for nucleocapsid. Scale bar, 100 μm. Quantification of the viral infection percentage is shown in Fig 2C. (B) Quantification of the viral infection percentage in the presence of indicated chemicals with different concentrations of A, with the control normalized to 100%. Data are shown as mean ± SD from 10 representative images. (C) Representative LDH assay of Huh7-ACE2 cells treated with the indicated molecules at the same concentrations as in Fig 2A and infected with SARS-CoV-2 (MOI = 1) for 24 h with 3 technical replicates from two independent experiments. (D) Validation of cell entry assay of 293T and 293T-ACE2 cells by infection of SARS-CoV-2 Spike pseudotyped lentiviruses. (E) Cell entry assay of 293T-ACE2 cells by SARS-CoV-2 spike pseudotyped lentivirus for 24 h in the presence of indicated molecules at the same concentrations as in Fig 2A. Data are shown as mean ± SD from three independent experiments. Statistical analyses are performed using One-way ANOVA. ***, p < 0.001, n.s., not significant. (F) Cell viability assay of 293T-ACE2 cells treated with indicated molecules for 24 h. (G) Immunoblots of cell lysates and media of Huh7-ACE2 cells incubated with or without SARS-CoV-2 (MOI = 3) for 12 h and treated with 5 μg/ml BFA for the indicated time points. (H-K) Quantification of intracellular spike protein (H), intracellular N protein (I), extracellular N protein (J), and medium/lysate ratio of N protein (K) in G. Data are presented as mean ± SD from 4 replicates of two independent experiments.
(TIF)

**S4 Fig. SARS-CoV-2 remodels the Golgi, and viral particles accumulate in Golgi fragments.** A gallery of EM images of Huh7-ACE2 cells incubated with or without SARS-CoV-2 (MOI = 1) for 24 h under two different magnifications. Boxed areas on the left images are enlarged and shown on the right. Scale bars, 500 nm.
(TIF)

**S5 Fig. SARS-CoV-2 infection alters the Golgi structure and GRASP55 and TGN46 protein level.** (A) Representative confocal images of Huh7-ACE2 cells incubated with or without SARS-CoV-2 (MOI = 1) for 24 h and stained for a

trans-Golgi marker GCC88 and nucleocapsid. (B-E) Quantification of A for the percentage of cells with fragmented Golgi (B), GCC88 item number (C), area (D), and relative expression level (E). (F) Representative confocal images of Huh7-ACE2 cells incubated with or without SARS-CoV-2 (MOI = 1) for 24 h and stained for a trans-Golgi marker Arl1 and nucleocapsid. Boxed areas in the upper panels of A and F are enlarged and shown underneath. Scale bars, 10 µm. (G-J) Quantification of Arl1 in F. (K) Immunoblots of indicated proteins in Vero E6 cells incubated with or without SARS-CoV-2 (MOI = 1) for 24 h. (L-Q) Quantification of K for the relative level of GRASP55 (L), GRASP65 (M), TGN46 (N), cathepsin D (CatD, O), EEA1 (P), and GPP130 (Q). (R) Representative confocal images of Calu-3 cells incubated with or without SARS-CoV-2 (MOI = 1) for 24 h and stained for a medial/trans-Golgi marker GRASP55 and nucleocapsid. (S-T) Quantification of R for the GRASP55 item number (S) and relative expression level (T). (U) Representative confocal images of Calu-3 cells incubated with or without SARS-CoV-2 (MOI = 1) for 24 h and stained for a TGN marker TGN46 and nucleocapsid. (V-W) Quantification of R for the TGN46 item number per cell (V) and relative expression level (W). Quantitation data are shown as mean ± SD from at least three independent experiments. Statistical analyses were performed using two-tailed Student's t-test. **, p < 0.01; ***, p < 0.001; n.s., not significant. (TIF)

**S6 Fig. GRASP55 expression reduces spike trafficking.** (A) Representative confocal images of stable 293T-ACE2 cells expressing GFP or GRASP55-GFP. Scale bar, 100 µm. (B) Cell entry assay of 293T-ACE2 cells stably expressing either GFP or GRASP55-GFP by SARS-CoV-2 spike pseudotyped lentivirus. (C) GRASP55 expression reduced spike at the cell surface. Huh7-ACE2 cells were transfected with GFP or GRASP55-GFP for 24 h, and then co-transfected with GFP or GRASP55-GFP together with spike-strep for 24 h. Cell surface proteins were biotinylated, pulled down by streptavidin beads, and blotted for indicated proteins. (D) Quantification of the Golgi item number of GCC88 of Huh7-ACE2 cells stably expressing GFP or GRASP55 infected by SARS-CoV-2 for 5, 8, and 24 h. Data are shown as mean ± SD from more than 24 random images from two independent experiments. (E) Representative confocal images of Huh7-ACE2 cells stably expressing GFP or GRASP55-GFP infected by SARS-CoV-2 for 24 h. Scale bar, 10 µm. (F) Pearson correlation coefficient between spike and GM130 in E. Statistical analyses were performed using two-tailed Student's t-test. ***, p < 0.001; n.s., not significant. (TIF)

**S7 Fig. GRASP55 depletion increases viral infection at late but not early stage.** (A) Representative confocal images of Huh7-ACE2 cells transfected with siControl or siGRASP55 oligos for 48 h followed by infection with SARS-CoV-2 for 5, 8, and 24 h. Scale bar, 100 µm. (B) Quantification of the viral infection percentage in B. Data are shown as mean ± SD from 30 random images from two independent experiments. Statistical analyses were performed using One-way ANOVA. ***, p < 0.001; n.s., not significant. (C) GRASP55 depletion increased ACE2 level at the cell surface. Huh7-ACE2 cells were transfected with siControl or siGRASP55 RNAi oligoes for 72 h. Cell surface proteins were biotinylated, pulled down by streptavidin beads, and blotted for indicated proteins. (D) Representative confocal images of Huh7-ACE2 cells transfected with siControl and siGRASP55 RNAi oligoes. Scale bar, 10 µm. (TIF)

**S8 Fig. Depletion of TGN46 inhibits SARS-Co-V secretion without affecting viral entry.** (A) Cell entry assay of 293T-ACE2 cells transfected with siControl or siTGN46 oligos for 48 h followed by infection with SARS-CoV-2 spike pseudotyped lentivirus for 24 h. Statistical analyses were performed using two-tailed Student's t-test. n.s., not significant. (B) Immunoblots of cell lysates of 293T-ACE2 cells transfected with siControl or siTGN46 oligos for 48 h followed by infection with SARS-CoV-2 spike pseudotyped lentivirus for 24 h. (C) TGN46 depletion reduces spike protein at the cell surface. Huh7-ACE2 cells were transfected with siControl or siTGN46 oligos for 48 h followed by transfection with spike-strep for 24 h. Cell surface proteins were biotinylated, pulled down by streptavidin beads, and blotted for indicated proteins. Note

that TGN46 level is enhanced at the cell surface by spike protein expression. (D) Representative confocal images of Huh7-ACE2 cells stably expressing mCherry and TGN46-mCherry. Scale bar, 10 µm. (E) Proposed working model for a novel role of TGN46 in SARS-CoV-2 infection. In brief, under normal conditions (a) TGN46 is expressed at a relatively low level and recycles between the Golgi and plasma membrane. After SARS-CoV-2 infection (b), TGN46 is up-regulated, which accelerates viral trafficking. When TGN46 is depleted (c), the trafficking speed of all variants of SARS-CoV-2 is reduced. Thus, TGN46 may serve as a carrier for viral trafficking and release. Created with BioRender.
(TIF)

**S1 Video. Intact Golgi in a control cell.** 3D reconstruction of Huh7-ACE2 cells stained for GM130. Images were taken for a total of 40 stacks and maximum intensity projection was performed.
(MP4)

**S2 Video. SARS-CoV-2 infection causes Golgi fragmentation.** Huh7-ACE2 cells were infected with SARS-CoV-2 (MOI = 1) for 24 h, fixed by 4% PFA, and stained for spike and GM130. Images were taken for a total of 40 stacks and maximum intensity projection was performed. Note the colocalization of spike and GM130 in the Golgi fragments.
(MP4)

**S1 Table. Antibody list used in this study.**
(XLSX)

**S2 Table. qPCR primers used in this study.**
(XLSX)

## Acknowledgments

We thank Dr. Nevan Krogan for kind gifts of plasmids encoding Strep-tagged SARS-CoV-2 proteins; Drs. Zhe Han, Jin-Gu Lee, Weichao Zhang, Ming Li, Meiqin Hu, and Haoxing Xu for reagents; and Dr. Gregg Sobocinski for technical assistance on electron microscopy. We thank Drs. Christiane Wobus, Akira Ono, Lois Weisman, Kenneth Cadigan, Mohammed Akaaboune, Ming Li, and current and past members of the Wang lab (especially Erpan Ahat and Jie Li) and the Ginsburg lab, for stimulating discussions and suggestions.

## Author contributions

**Conceptualization:** Jianchao Zhang, David Ginsburg, Andrew W. Tai, Yanzhuang Wang.

**Formal analysis:** Jianchao Zhang.

**Funding acquisition:** David Ginsburg, Andrew W. Tai, Yanzhuang Wang.

**Investigation:** Jianchao Zhang, Andrew Kennedy, Daniel Macedo de Melo Jorge, Lijuan Xing, Whitney Reid, Sarah Bui, Joseph Joppich, Molly Rose, Sevval Ercan.

**Methodology:** Jianchao Zhang, Andrew Kennedy, Daniel Macedo de Melo Jorge.

**Project administration:** Jianchao Zhang, Andrew W. Tai, Yanzhuang Wang.

**Resources:** Qiyi Tang, David Ginsburg, Andrew W. Tai, Yanzhuang Wang.

**Supervision:** Jianchao Zhang, Andrew W. Tai, Yanzhuang Wang.

**Validation:** Jianchao Zhang.

**Visualization:** Jianchao Zhang.

**Writing – original draft:** Jianchao Zhang, Andrew W. Tai, Yanzhuang Wang.

**Writing – review & editing:** Jianchao Zhang, Andrew Kennedy, Daniel Macedo de Melo Jorge, Lijuan Xing, Whitney Reid, Sarah Bui, Joseph Joppich, Molly Rose, Sevval Ercan, Qiyi Tang, David Ginsburg, Andrew W. Tai, Yanzhuang Wang.

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
