## [Decision Letter · Decision Letter 0]

PPATHOGENS-D-25-00934

SARS-CoV-2 remodels the Golgi apparatus to facilitate viral assembly and secretion

PLOS Pathogens

Dear Dr. Wang,

Thank you for submitting your manuscript to PLOS Pathogens. According to the review comments, we invite you to submit a revised version of the manuscript that addresses the points raised during the review process.

Please submit your revised manuscript within 60 days . If you will need more time than this to complete your revisions, please reply to this message or contact the journal office at plospathogens@plos.org. Please include the following items when submitting your revised manuscript:

We look forward to receiving your revised manuscript.

Kind regards,

Ke Peng

Academic Editor

PLOS Pathogens

Alexander Gorbalenya

Section Editor

PLOS Pathogens

 Sumita Bhaduri-McIntosh

Editor-in-Chief

PLOS Pathogens

orcid.org/0000-0003-2946-9497 Michael Malim

Editor-in-Chief

PLOS Pathogens

orcid.org/0000-0002-7699-2064

**Additional Editor Comments :**

Dear Dr. Wang,

Thank you very much for submitting your manuscript " SARS-CoV-2 remodels the Golgi apparatus to facilitate viral assembly and secretion" for consideration at PLOS Pathogens. As with all papers reviewed by the journal, your manuscript was reviewed by members of the editorial board and by several independent reviewers. In light of the reviews (below this email), we would like to invite the resubmission of a revised version that takes into account the reviewers' comments, particularly comments from Reviewer 3.

Sincerely,

**Journal Requirements:**

https://journals.plos.org/plospathogens/s/submission-guidelines#loc-parts-of-a-submission

3) Thank you for stating " USA-WA1/2020 was isolated from an oropharyngeal swab from a patient who returned from an affected region of China and developed COVID-19 on January 19, 2020, in Washington, USA."

Please insert an Ethics Statement at the beginning of your Methods section, under a subheading 'Ethics Statement'. It must include:

i) The full name(s) of the Institutional Review Board(s) or Ethics Committee(s)

ii) The approval number(s), or a statement that approval was granted by the named board(s)

iii) A statement that formal consent was obtained (must state whether verbal/written) OR the reason consent was not obtained (e.g. anonymity). NOTE: If child participants, the statement must declare that formal consent was obtained from the parent/guardian.].

5) We notice that your supplementary Figures, and Tables are included in the manuscript file. Please remove them and upload them with the file type 'Supporting Information'. Please ensure that each Supporting Information file has a legend listed in the manuscript after the references list.

Potential Copyright Issues:

i) Figures 5N, and S8E: We note that the figures are created through BioRender. Please confirm that you hold a Premium account and provide a pdf copy of the CC BY 4.0 Licence as provided by BioRender. For instructions on how to generate a CC BY 4.0 license for your figure, please see the guidelines here: https://help.biorender.com/hc/en-gb/articles/21282341238045-Publishing-in-open-access-resources. 

If you are using the free assets from BioRender, we are unable to publish these images as they are licenced under a stricter licence than CC BY 4.0. In this case we ask you to remove the BioRender images and replace them with open source alternatives.

See these open source resources you may use to replace images / clip-art:

- https://bioart.niaid.nih.gov/ 

- https://bioicons.com/

- https://healthicons.org/ 

- https://scidraw.io/

- https://reactome.org/icon-lib

- https://www.phylopic.org/images

7) In the online submission form, you indicated that "All data can be provided upon request if the manuscript is accepted for publication." All PLOS journals now require all data underlying the findings described in their manuscript to be freely available to other researchers, either

1. In a public repository

2. Within the manuscript itself

3. Uploaded as supplementary information.

8) Please amend your detailed Financial Disclosure statement. This is published with the article. It must therefore be completed in full sentences and contain the exact wording you wish to be published.

2) If any authors received a salary from any of your funders, please state which authors and which funders.

9) Please ensure that the funders and grant numbers match between the Financial Disclosure field and the Funding Information tab in your submission form. Note that the funders must be provided in the same order in both places as well. Currently, the order of this grant "R35HL171421" is different in both places.

**Reviewers' Comments:**

Reviewer's Responses to Questions

**Part I - Summary**

Reviewer #1: The authors have addressed all my comments. I am positively impressed by the quality of the revised manuscript. I do not have any additional concern.

Reviewer #2: Zhang et al. demonstrated in this study that the Golgi apparatus and many other organelles are

disturbed by SARS-CoV-2 infection. They focused on the Golgi apparatus and especially on

TGN46 and GRASP55 which are both affected differently in their level of expression by the

SARS-CoV-2 infection. TGN46 is overexpressed while GRASP55 is decreased in expression.

The authors nicely demonstrated that modulation of both proteins either increased or decreased particles production. They

demonstrated that particle secretion was impacted in absence of TGN46 suggesting that SARS-CoV2 divert specific host pathways for its release at the plasma membrane. GRASP55 depletion at the contrary, increases viral secretion, although it is still unclear how the fragmentation of the Golgi apparatus could play a role in increasing particle secretion. The study is interesting and highlight the involvement of the Golgi apparatus in SARS-CoV2 life cycle with key players identified.

Reviewer #3: SARS-CoV-2 remodels the Golgi apparatus to facilitate viral assembly and secretion

This submission documents Golgi fragmentation in SARS-CoV-2 infected cells, thereby confirming previous reports. The study proceeds by treating infected cells with various chemicals that are known to perturb protein synthesis and trafficking, with focus on BFA, a drug that collapses Golgi and has been tested for inhibition of coronavirus MHV secretion. Unlike MHV, BFA blocked SARS-CoV-2 infection. There are then EM images that are used to support statements that SARS-CoV-2 particles reside in Golgi vesicles. The works then advance to GRASP55 and TGN46 because these two Golgi proteins were discovered as reduced and elevated, respectively, in IFA images of infected cells. GRASP55 was then forcibly increased or decreased by transgene expression or siRNA KD. TGN46 was decreased by siRNA KD. Increased GRASP55 did modestly reduce SARS-CoV-2 infection while decreased GRASP55 increased infection. Decreased TGN46 modestly decreased SARS-CoV-2 infection. From these findings, the authors primary conclusion is that the Golgi is required for SARS-CoV-2 secretion.

The results are interesting and potentially valuable because they address current questions about the morphogenesis and egress of coronaviruses particles from infected cells. This study adds some insights into the possible roles for Golgi vesicles in virus secretion. While the authors have responded to three “review commons” recommendations, there are still several concerns with some data interpretations and conclusions. There are also some suggestions for manuscript modification.

**Part II – Major Issues: Key Experiments Required for Acceptance**

Reviewer #1: (No Response)

Reviewer #2: The study was nicely improved from the first version of the manuscript. Most of the experiments suggested, were performed and carefully done. The authors supported their microscopy observations with RT-qPCR, confirmed that their observations were not caused by cell death using LDH assays. The biochemical studies to observe particles secretion and the disturbance caused by the virus in host cell receptor localization at the plasma membrane was convincing.

Reviewer #3: Comments:

1. Much of the report seems to be framed around a prior publication in which secretion of the coronavirus MHV was reported to bypass the conventional Golgi secretory pathway (Cell 183: 1520-1535 e1514). Given this perspective, it would seem that SARS-CoV-2 and MHV should be compared directly. Huh-7 – MHV receptor-positive cells can be infected by MHV.

2. While data do show that GRASP55 increases/decreases will decrease/increase SARS-CoV-2, the effects appear modest. Similarly, decreased TGN46 appears to modestly decrease SARS-CoV-2 infection and in the TGN46 case, this reviewer sees little evidence that the effect is at the level of virus secretion. The data in Figs 5, 6 and 7 should be plotted differently, specifically focusing on viral proteins and infectious virions in media vs. lysates. Plotting media/lysate ratios in the case of TGN46 KD might show that TGN46 is reducing infection at stages that are not related to assembled virion secretion (in particular, Fig 7F and 7OP data are not convincingly showing that TGN46 KD impairs virus secretion). Plotting media/lysate ratios for S and N proteins may more convincingly help to determine whether GRASP55 operates at the level of spike protein incorporation into virus particles. Furthermore, M proteins (not N proteins) are the principal coronavirus proteins that operate in virus assembly and the S and M protein ratios may be more illuminating than S and N ratios.

3. The word “expression” is used throughout the paper in reference to changes in GRASP55 and TGN46 protein levels in virus-infected cells. This gives the impression that virus infections are altering GRASP55 and TGN46 gene expressions to facilitate virus assembly and secretion, but gene expressions were not tested after virus infection. Might the altered GRASP55 and TGN46 levels be due to altered protein stabilities following infection, perhaps due to prior Golgi fragmentations? If this remains a possibility, it should be noted and text changed accordingly.

Minor comments:

1. Figs 1E-H; is GPP130 or GRASP65 being imaged? (legend says GRASP65)

2. Figs 2AB; infection “rates” not measured, rather, percent infected cells are measured – consider rephrasing text and ordinat axes

3. Figs 2AB; should note that many of the chemical conditions lack controls (cannot be certain whether some chemicals are active under any condition). Consider removing the data on those drugs that were found to have no effect on infection and were also not shown to have any effects on events in which they should be effecting changes.

4. Fig 2D; should have a positive control condition for LDH release (this applies to all of the data using LDH release as a measure of compromised cells)

5. Fig 2E; should show intracellular N in comparison to the N in media

6. Fig 2C-G; should include MHV-infected cells as comparative control, especially given the importance of this finding and given that the text highlights potential distinctions between SARS-CoV-2 and the previous findings reporting MHV resistance to BFA.

7. Fig 3A-E; is GRASP65 or GM130 being imaged? (legend says GM130)

8. Fig 3P; can agree that white arrowheads are probably pointing to viral particles, but how can it be claimed that black arrows and arrowheads are pointing to golgi membranes and DMVs, respectively?

9. Fig 4; “down and up-regulate” may not be best terms for results that show host protein levels within infected cells by WB (see major comments)

10. Fig 6; the word “accelerates” in the Fig 6 titles is not appropriate, given that kinetic analyses of virus assembly and secretion were not performed.

11. Fig 7; the word “required” in the Fig 7 title is not appropriate, given the modest effects of TGN46 KD on SARS-CoV-2 infection.

12. Discussion, line 460; “integrity of conventional trafficking pathway is essential for SARS-CoV-2..” This statement is puzzling because the data show that the conventional trafficking pathway is disturbed by Golgi fragmentation.

**Part III – Minor Issues: Editorial and Data Presentation Modifications**

Reviewer #1: (No Response)

Reviewer #2: No minor modifications

Reviewer #3: (No Response)

PLOS authors have the option to publish the peer review history of their article (what does this mean? ). If published, this will include your full peer review and any attached files.

**Do you want your identity to be public for this peer review?** For information about this choice, including consent withdrawal, please see our Privacy Policy .

Reviewer #1: No

Reviewer #2: No

Reviewer #3: No

**Figure resubmission:**
---

## [Decision Letter · Decision Letter 1]

PPATHOGENS-D-25-00934R1

SARS-CoV-2 remodels the Golgi apparatus to facilitate viral assembly and secretion

PLOS Pathogens

Dear Dr. Wang,

Thank you for submitting your manuscript to PLOS Pathogens. We invite you to submit a revised version of the manuscript that addresses the points raised during the review process.

Please submit your revised manuscript within 30 days Aug 03 2025 11:59PM. If you will need more time than this to complete your revisions, please reply to this message or contact the journal office at plospathogens@plos.org. Please include the following items when submitting your revised manuscript:

We look forward to receiving your revised manuscript.

Kind regards,

Ke Peng

Academic Editor

PLOS Pathogens

Alexander Gorbalenya

Section Editor

PLOS Pathogens

Sumita Bhaduri-McIntosh

Editor-in-Chief

PLOS Pathogens

orcid.org/0000-0003-2946-9497

Michael Malim

Editor-in-Chief

PLOS Pathogens

orcid.org/0000-0002-7699-2064

**Additional Editor Comments :**

Please replace "Nsps" with "nsps" per standard for coronaviruses (consult doi.org/10.1038/s41564-020-0695-z) 

**Journal Requirements:**

1) Please ensure that the funders and grant numbers match between the Financial Disclosure field and the Funding Information tab in your submission form. Note that the funders must be provided in the same order in both places as well. Currently, the order and the recipient of this grant "R35HL171421" is different in both places. Please also ensure that the funders of these grants "R21AI152865" and "R01NS102279" match in both places. 

**Reviewers' Comments:**

Reviewer's Responses to Questions

**Part I - Summary**

Reviewer #3: Thank you for responses to my review. While the abundant data, the careful experimentation, and the important topic are all very impressive, I remain skeptical of some data interpretations. Note, for example, the reviewer 3 response #5 that addresses effects of BFA on viral protein levels. The response includes a very clean WB showing that BFA dramatically reduces intracellular N proteins. Therefore it is not surprising that there is less secreted virus in response to BFA. Two points here; **first, the WB that is presented in responses to reviewers should be in the paper** . Second, perhaps more to the point; the overall findings point to BFA blocking N translation or stability, making it hard to interpret findings in context of virus assembly. The authors claim that BFA is blocking virus "assembly" but the data appear to indicate that the BFA effect is at some earlier infection stage.

The above findings related to BFA and effects on N proteins relates to this reviewer's separate suggestion to plot data in terms of media to lysate (M/L) ratios. If this were done for N proteins, i.e., in Fig 2 (and with the additional WB that was provided in response to reviewers), then the M/L ratios would suggest that BFA is not changing M/L ratios and therefore is not directly affecting virus secretion. Hence I stick with the suggestion that M/L ratios have value in data interpretation. The authors note correctly that both unassembled and virus-assembled proteins are present in lysates, while media contain mostly virus-assembled proteins. However, this is not really a confounder in analysis. If GRASP55 and TGN46 are directly affecting virus assembly and secretion, and not other infection stages, then they would be expected to change viral protein M/L ratios. I think the M/L ratios have value in data interpretation.

**Part II – Major Issues: Key Experiments Required for Acceptance**

Reviewer #3: (No Response)

**Part III – Minor Issues: Editorial and Data Presentation Modifications**

Reviewer #3: (No Response)

PLOS authors have the option to publish the peer review history of their article (what does this mean? ). If published, this will include your full peer review and any attached files.

**Do you want your identity to be public for this peer review?** For information about this choice, including consent withdrawal, please see our Privacy Policy .

Reviewer #3: No

**Figure resubmission:**
---

## [Editor Report · Decision Letter 2]

Dear Dr.Wang,

We are pleased to inform you that your manuscript 'SARS-CoV-2 remodels the Golgi apparatus to facilitate viral assembly and secretion' has been provisionally accepted for publication in PLOS Pathogens.

Best regards,

Ke Peng

Academic Editor

PLOS Pathogens

Alexander Gorbalenya

Section Editor

PLOS Pathogens

Sumita Bhaduri-McIntosh

Editor-in-Chief

PLOS Pathogens

orcid.org/0000-0003-2946-9497

Michael Malim

Editor-in-Chief

PLOS Pathogens

orcid.org/0000-0002-7699-2064
---

## [Editor Report · Acceptance letter]

Dear Dr. Wang,

We are delighted to inform you that your manuscript, "SARS-CoV-2 remodels the Golgi apparatus to facilitate viral assembly and secretion," has been formally accepted for publication in PLOS Pathogens.

Best regards,

Sumita Bhaduri-McIntosh

Editor-in-Chief

PLOS Pathogens

orcid.org/0000-0003-2946-9497

Michael Malim

Editor-in-Chief

PLOS Pathogens

orcid.org/0000-0002-7699-2064